# Advancements in Plasma Agriculture: A Review of Recent Studies

**DOI:** 10.3390/ijms242015093

**Published:** 2023-10-11

**Authors:** Evgeny M. Konchekov, Namik Gusein-zade, Dmitriy E. Burmistrov, Leonid V. Kolik, Alexey S. Dorokhov, Andrey Yu. Izmailov, Babak Shokri, Sergey V. Gudkov

**Affiliations:** 1Prokhorov General Physics Institute of the Russian Academy of Sciences, 119991 Moscow, Russia; ngus@mail.ru (N.G.-z.); dmitriiburmistroff@gmail.com (D.E.B.); leonidkolik@mail.ru (L.V.K.); s_makariy@rambler.ru (S.V.G.); 2Federal Scientific Agroengineering Center VIM, 109428 Moscow, Russia; dorokhov@rgau-msha.ru (A.S.D.);; 3Physics Department, Shahid Beheshti University, Tehran 1983969411, Iran

**Keywords:** plants, plasma source, plasma-activated water, reactive oxygen species, reactive nitrogen species, dielectric barrier discharge, plasma jet, corona discharge, spark discharge, germination

## Abstract

This review is devoted to a topic of high interest in recent times—the use of plasma technologies in agriculture. The increased attention to these studies is primarily due to the demand for the intensification of food production and, at the same time, the request to reduce the use of pesticides. We analyzed publications, focusing on research conducted in the last 3 years, to identify the main achievements of plasma agrotechnologies and key obstacles to their widespread implementation in practice. We considered the main types of plasma sources used in this area, their advantages and limitations, which determine the areas of application. We also considered the use of plasma-activated liquids and the efficiency of their production by various types of plasma sources.

## 1. Introduction

The unique properties of physical plasma and its ability to operate at atmospheric pressure make it an attractive technology for numerous scientific and industrial applications, ranging from medicine and agriculture to electronics and materials science [1]. For example, this technology has proven to be a simple and low-cost approach for nanoparticle synthesis [2] or an effective surface modification agent to produce superhydrophobic and superoleophilic films for oil-water separation and self-cleaning [3]. In medicine, plasma is widely used to solve problems of creating biocompatible materials, orthopedics [4], oncology, dentistry [5], dermatology [6], etc.

In recent years, plasma agriculture has emerged as a promising and innovative field with the potential to evaluate conventional farming practices. The application of plasma-based technologies in agriculture offers new opportunities to address various challenges faced by the agricultural industry, such as food security, environmental sustainability, and the need for increased crop yields [7].

The generation of plasma can be achieved through various methods. Each of these techniques offers distinct advantages and can be tailored to suit specific agricultural needs. Moreover, the use of plasma-activated water (PAW) has emerged as a promising technology with potential applications in agriculture. PAW is produced by subjecting water to discharges, resulting in the generation of various active species, including reactive oxygen (ROS) and nitrogen species (RNS) [8,9].

One of the key areas of focus in plasma agriculture is plant growth promotion. Plasma and PAW treatments have been shown to stimulate seed germination, enhance root development, and increase overall crop productivity. These effects are attributed to the activation of various plant growth regulators, modification of plant hormone levels, and the induction of stress tolerance mechanisms. Furthermore, plasma can also improve nutrient availability and uptake by altering the physicochemical properties of the soil, leading to increased nutrient-use efficiency and reduced fertilizer requirements.

Another aspect of plasma agriculture is its potential for pest and disease management. Plasma treatments have demonstrated efficacy in controlling pathogens, offering an environmentally friendly alternative to traditional chemical-based pesticides. The antimicrobial properties of plasma, coupled with its ability to induce systemic resistance in plants, provide an approach to disease prevention and control.

Furthermore, plasma and PAW treatments have been shown to alleviate the detrimental effects of abiotic stress factors such as drought, salinity, and extreme temperatures. By activating stress-responsive genes and enhancing antioxidant defense mechanisms, plasma can improve plant resilience and ensure sustainable crop production under challenging environmental conditions.

This review paper aims to provide an overview of recent achievements in plasma agriculture, highlighting the significant advancements made in the field. By examining a comprehensive range of studies conducted in the past few years, we aim to present an up-to-date perspective on the potential of plasma-based technologies in addressing plant development studies.

## 2. Physical Plasma and Its Effect on Objects

Physical plasma is a partially or fully ionized gas that has the properties of quasi-neutrality and collectivity. The interaction of plasma with an object is complex and highly dependent on various factors, such as the plasma’s characteristics, the nature of the object’s surface, and the specific application or purpose of the plasma treatment. Plasma finds applications in a very wide range of fields (Figure 1), including controlled fusion, environmental remediation [10,11,12,13], surface modification [14,15], modification of petroleum products [16,17], synthesis of micro- [18,19] and nanomaterials [20,21], disinfection [22,23,24], medicine [25,26,27,28,29,30,31,32], agriculture [33,34,35,36], food processing [37,38] etc.

When directed towards biological materials, such as cells, tissues, or microorganisms, plasma can induce a range of effects with potential applications:Plasma contains high-energy electrons that can collide with biological objects, leading to energy transfer, excitation, and ionization of molecules. These processes can induce various chemical reactions within the biological material.The presence of charged particles in plasma generates electric fields that can influence the behavior of charged molecules and ions within the biological object.Plasma produces a multitude of reactive species, including free radicals (e.g., OH•, O•, N•), ions, and excited molecules. These species play a significant role in initiating chemical reactions within the biological material.The reactive oxygen and nitrogen species generated by plasma can lead to oxidative stress, affecting cellular components and signaling pathways.Plasma can modify the surface properties of materials, altering characteristics such as roughness, wettability, and chemical composition [39]. These modifications can influence cellular adhesion and interactions [40].

Achieving the desired outcomes requires optimizing plasma parameters. Tailoring these parameters can influence the balance between beneficial effects and potential harm. To minimize the negative effects of plasma exposure on sensitive objects, plasma-activated water (PAW) or plasma-treated solutions (PTS) can be used as a treatment agent. PAW, among all the influencing factors, is restricted to the action of ROS and RNS, as well as other chemical compounds (Figure 2). However, research indicates that selecting concentrations of bioactive compounds makes it possible to achieve effects similar to those obtained through direct plasma treatment. Hence, the creation of PAW and PTS with specific compositional characteristics for various biological and technical applications has emerged as a distinct scientific field.

In practice, determining the optimal parameters for plasma-liquid interaction can be quite challenging. These parameters include the applied voltage, discharge gas flow rate, treatment duration, distance of the electrode from the solution surface, and the potential concentration of precursors or reducing agents. In the gas and liquid phases, free electrons receive energy from the electric field created between electrodes. This energy leads to the generation of relatively high-energy electrons. These high-energy electrons can then interact with various components, such as oxygen, water molecules, and other ionic compounds (**1**–**3**), triggering a range of physical and chemical reactions. These processes have the potential to produce reactive chemical species such as OH•, O•, H•, ONOO^−^, NO•, H_2_O_2_, and ultraviolet (UV) radiation. These species can be detected through the emission of photons, which can be analyzed using optical emission spectroscopy (OES).
N_2_ + e^−^ → N• + N• + e^−^,(1)
O_2_ + e^−^ → O• + O• + e^−^,(2)
H_2_O + e^−^ → H• + OH• + e^−^.(3)

Nitrogen in the emission spectrum can be relatively clearly determined from the N_2_ s positive system (C^3^Π_u_−B^3^Π_g_) at a wavelength from 315 to 433 nm. Refs [3,41]. The presence of O_2_ in the discharge zone can be indicated by the NO group peaks from 214 to 270 nm. The •HO radicals are confirmed by the peak at about 296 nm or 310 nm and the oxygen radical at 284 nm [42]. The emission peaks corresponding to •OH and •O may be attributed to the fragmentation of H_2_O molecules. These H_2_O molecules can either diffuse in from the surrounding ambient air or evaporate from the liquid’s surface.

To achieve the required biological effect, it is also important to analyze nanoparticles (NP) that can form in plasma-activated water as a result of erosion of electrodes or other metal objects immersed in the liquid. Transmission electron microscopy can be used to obtain surface morphology and particle size distribution, whereas elemental characterization of NP is often obtained with energy-dispersive spectroscopy. The stability test of NP and elemental characterization of PAW are analyzed, for example, using ultraviolet–visible spectroscopy (UV–Vis). Thus, the UV–Vis absorption peaks at around 410 nm correspond to the silver NP formation [2].

In this review, we will explore the results of studies focusing on the effects of both direct plasma treatment and indirect treatment (using PAW) on plants, aiming to investigate plant physiology and enhance agricultural intensification. Special attention will be given to research conducted in recent years, and we will categorize these studies based on the type of plasma source employed.

## 3. Plasma Sources in Biological Sciences and Agriculture

Plasma sources used in plasma agriculture can vary in design and operation, but their primary goal is to generate and deliver plasma to plants, seeds, soil, or water in controlled and targeted ways. They may be incorporated into portable devices, hand-held tools, or larger systems designed for agricultural operations, depending on the scale and requirements of the application. The plasma used in such applications is commonly referred to as non-thermal (NTP) or cold atmospheric plasma (CAP) because during processing, there is no significant heating of the object. NTP sources (Figure 3) typically use electrical discharges, such as:dielectric barrier discharge (DBD),corona discharge,spark discharge,atmospheric pressure plasma jets (APPJ) and plasma torch,underwater discharge.

### 3.1. Dielectric Barrier Discharge

DBD occurs when a high voltage is applied to a gas-filled gap between two dielectric materials or applied to one of the parallel electrodes separated by a dielectric barrier, which acts as an insulating layer between them. The dielectric barrier prevents a direct current flow between the electrodes while allowing the alternate or pulsed electric field to penetrate the gap. As a result, the electric field induces the ionization of the gas molecules in the vicinity of the dielectric surface, creating a plasma region. The gradients of the electric field lead to the formation of filamentous micro-discharges, which rapidly propagate across the discharge gap. These micro-discharges are responsible for generating a significant number of energetic electrons and ions. DBD is mainly used in applications where thermal damage needs to be minimized. Additionally, due to relative ease in scaling, DBD has been employed in surface treatment processes, such as disinfection, improving adhesion properties, modifying surface wettability, or promoting the deposition of thin films [43].

### 3.2. Corona Discharge

Corona discharge represents an electrical phenomenon wherein partial ionization of the gas surrounding a conductor occurs due to the influence of a high electric field. It is characterized by the formation of a luminous crown-like glow near the conductor. This occurrence is typically observed when the magnitude of the electric field gradient or potential difference between the conductor and its surrounding medium, typically air, surpasses a specific threshold referred to as the breakdown voltage. Once this threshold is exceeded, electrons within the gas molecules in the vicinity of the conductor acquire sufficient energy to initiate ionization processes. Consequently, positive ions and free electrons are generated as a result. The interaction of ions and electrons gives rise to various consequential effects. For instance, the presence of ions can lead to the accumulation of charge on nearby objects or surfaces. The free electrons, on the other hand, partake in secondary processes, such as the collisional excitation of gas molecules, which can lead to ozone generation or the production of other reactive species [44,45].

Corona discharge is mainly used for the decontamination of the gaseous medium and treatment of liquids and soil surfaces [46,47,48].

### 3.3. Spark Discharge

Spark discharges occur when a high voltage is applied across the electrodes, creating a strong electric field in the gap. As the voltage increases, the electric field strength eventually exceeds the breakdown voltage of the gas, leading to ionization and the formation of a conductive plasma channel. This ionized channel allows the flow of electric current between the electrodes, resulting in a spark discharge. These discharges typically occur over a short duration, ranging from nanoseconds to milliseconds, depending on the power supplies, gap distance, and gas composition. Spark discharges have various practical agriculture applications. They are commonly observed in electrical systems, where they act as the initiator of the plasma jet. They can also be used for direct processing of surfaces that are not subject to significant changes under the influence of heat (e.g., PAW and PTS generation) [49,50,51,52,53]. To increase the efficiency of water treatment, multi-spark systems are used [54,55,56].

### 3.4. Atmospheric Pressure Plasma Jets and Plasma Torch

A plasma jet is a stream of plasma that is expelled from a nozzle or an electrode by subjecting a gas to an intense electric field. APPJs are characterized by a wide range of temperatures, depending on the specific plasma generation method and the gas composition. High-energy systems can be implemented using a microwave plasma torch. It employs microwave energy to excite and ionize the gas within a resonant cavity. The gas is introduced into the cavity, and the intense electromagnetic field of the microwaves interacts with the gas molecules, leading to plasma formation. On the other hand, low-energy APPJs can be generated using, e.g., spark discharges or DBD.

Microwave plasma torches are commonly used in material processing applications, chemical synthesis, environmental remediation, and PAW production [57,58,59,60,61,62]. Low-energy APPJs are utilized for precise surface treatments, cleaning, and activation, as well as in fields such as plasma medicine, including wound healing, sterilization, and cancer treatment [63,64,65].

### 3.5. Underwater Discharge

Underwater discharge is an electrical discharge that occurs in a submerged environment. It involves the flow of electric current through a conductive medium, leading to various effects such as electrolysis, heat generation, nanoparticle generation, and gas bubble formation. The specific characteristics and behavior of the discharge depend on factors such as the applied voltage, the conductivity of the medium, and the geometry of the electrodes. In agriculture, the underwater discharge is used primarily to purify water or produce PAW and PTS with the widest range of possible chemical composition [66].

## 4. Direct Plasma Treatment Results

To analyze the results, we divided the papers according to the type of plasma source used. An overview of direct plasma treatment results in recent years is presented in Table 1.

### 4.1. Dielectric Barrier Discharge

In the study [67], two different sample forms of soybeans (whole and crushed beans) were subjected to DBD plasma treatment using two different plasma chemistry modes (ozone and nitrogen oxides) and a novel pressure-swing reactor. The paper highlights the potential of plasma treatment in enhancing the protein extraction yield and modifying the functional properties of soybean proteins, particularly in crushed soybean samples.

The water absorption and germination of spinach seeds were enhanced after treatment with rollable DBD [68]. The DBD plasma treatment of sunflower seeds [69] led to the accelerated development and production of taller seedlings with higher total weight compared to untreated controls. Subsequent evaluation of field-grown mature plants indicated that plasma-treated seeds had a positive impact on head size, increasing the number of seeds per head and the weight per thousand seeds, resulting in a significant increase in overall yield. Furthermore, the beneficial effects of plasma treatment endured for at least two weeks of seed storage, indicating that the treatment’s positive influence on growth and yield persisted even after a storage period.

DBD plasma treatment has beneficial effects on lettuce [70], leading to increased yield, improved nutrient content, and potentially enhanced photosynthetic efficiency, making it a promising technique for enhancing lettuce production and quality.

Treatment at the seed stage led to alterations in certain biochemical components of wheat germ [71], including increased chlorophyll, flavonoids, and polyphenols. Additionally, the ash content in plasma-treated wheat varieties was higher compared to the untreated ones, while other nutritional parameters remained relatively unchanged. These findings may have implications for improving the nutritional value and potential health benefits of wheat germ in food applications. The use of a coaxial DBD for treating winter wheat seeds also showed promising results [90]. Both argon (Ar) and helium (He) plasma treatments resulted in a significant increase in the wettability of the seed surface and accelerated germination. A low-pressure DBD treatment [78] of wheat seeds led to surface oxidation, improved water absorption, and increased germination rate. The treatment induced the accumulation of hydrogen peroxide, enhanced expression of antioxidant genes, and positively influenced grain yield. The higher iron and fat content in the grains, coupled with reduced moisture content, could further enhance their nutritional value and storability.

Despite the initial decrease in seedling emergence, plasma treatment proved beneficial for the growth, yield, and biochemical composition of buckwheat seeds [85].

A positive impact was registered on watercress seeds [72], and significant growth stimulation was promoted in the resulting seedlings. The ability of the active species to penetrate inside the seeds suggests a complex mechanism of action, potentially involving internal biological processes that contribute to the observed growth enhancement.

The study [73] showed that DBD plasma treatment significantly improved eggplant (*Solanum melangena* L.) seed germination. Thus, treatment enhanced the antioxidant activity of the plants, indicating an increase in their ability to resist oxidative stress. Additionally, the higher levels of soluble sugars and proteins in the plants were observed, indicating improved nutrient availability and metabolism. The concentration of total phenols, which are known for their health-promoting properties, was also increased in the treated plants. Regarding mineral content, the CAP treatment led to an increase in the concentrations of essential minerals such as calcium (Ca), copper (Cu), iron (Fe), manganese (Mn), and potassium (K), indicating improved nutrient uptake and assimilation by the plants. However, the content of zinc (Zn) in the treated plants was reduced compared to the control group, which may have implications for the overall nutrient balance and plant health.

For the white radish (*Raphanus sativus*) samples [74], the treatment also positively influenced the seed coat, making it more hydrophilic and enhancing water permeability. This likely contributed to improved germination and growth-related parameters in the seedlings, including increased vigor, chlorophyll content, and antioxidant activity. The treatment also enhanced the presence of beneficial compounds such as flavonoids and phenols in the seedlings, further supporting the overall growth and development of the plants.

DBD treatment proved to be highly beneficial in terms of inhibiting microbial growth [77], enhancing antioxidant activity, and preserving the overall quality and shelf life of *F. velutipes* mushrooms [75]. The treatment exhibited strong antimicrobial effects on mesophilic aerobic bacteria and filamentous fungi, causing severe damage to spores and reducing their viability [79]. Additionally, the treatment led to an increase in lycopene content in dried tomatoes without altering their antioxidant properties. However, the effects on seed germination and early seedling growth seem to be dependent on the duration of exposure. Prolonged exposure may have detrimental effects, possibly due to the disruption of cellular structures and processes caused by the generated reactive species [83]. Therefore, careful optimization of the treatment parameters is necessary to ensure desired outcomes and avoid potential negative effects on seeds and seedlings.

The importance of optimizing plasma treatment parameters to achieve the desired effects on seed viability and early seedling growth is described in [84]. Short-term plasma treatment appears to have a positive impact, while prolonged exposure, especially in a nitrogen environment, can be detrimental to seed germination. Understanding the effects of plasma treatment on seed physiology and DNA integrity is crucial for harnessing its potential benefits in agricultural practices.

The study [80] revealed that treatment time and stress level are essential parameters that stimulate the germination of *Arabidopsis thaliana* seeds when exposed to DBD plasma. Prolonged treatment times and higher stress levels were found to accelerate the germination process. The importance of considering specific seed characteristics, such as fatty acid composition, when determining the optimal plasma treatment time for achieving optimal germination is shown in [82]. The different responses observed in *A. thaliana* and *C. sativa* seeds are likely attributed to the contrasting changes in the fatty acid profiles induced by the plasma treatment, which may influence the neutralization of RONS and subsequent germination processes.

The increased levels of antioxidants and flavonoids could also enhance the nutritional value of the basil leaves [81]. Additionally, the plasma treatment’s ability to reduce microbial load may have practical applications in improving the safety and shelf life of basil products.

Plasma-mediated stimulation [86] during the early stages of pea seed development is positively correlated with an increase in ROS content, enhanced peroxidase activity in cell walls, and improved mechanical strength of the cell walls. These effects likely contributed to the higher resistance and vigor of the plasma-treated pea seedlings, leading to an increase in seed germination and early growth.

Paper [89] also showed that DBD can be used as a safe and effective method to enhance the germination and growth of pea seeds without causing significant DNA damage to the seedlings. However, it is important to note that the optimal exposure time for plasma treatment should be adhered to achieve the desired positive effects on seed germination and growth while minimizing any potential negative impacts on DNA integrity.

The impact on the activity of enzymes essential for the germination of pea seeds was studied in [91]. Infrared spectra analysis revealed that the plasma-generated reactive oxygen and nitrogen species, along with UV radiation, induced the oxidation of lipids and polysaccharides on the surface of the pea seeds, leading to increased wettability, which can facilitate germination.

The barley grains were treated in [87,92]. The results suggest that DBD shows promise as a potential method for reducing mycotoxin contamination without negatively affecting their nutritional quality. However, it can have both beneficial and harmful effects, depending on the exposure time and the surrounding atmosphere (ambient air, nitrogen, or oxygen). Short exposure times can enhance germination and enzyme activity, while longer exposure times can induce DNA damage, leading to genotoxic effects.

The study on the germination improvement of three pine species (*Pinus*) [88] showed that there were no statistically significant differences between the plasma-treated and untreated samples. However, there was a positive trend in the germination and growth of pine seeds, particularly after a short treatment duration of 3 s. This indicates that a brief exposure to plasma had a beneficial effect on the germination and initial growth of the pine seeds. On the other hand, when the exposure time to plasma was extended to 10 s or more, it had a retarding effect on germination and growth.

DBD treatment can effectively improve the viscoelastic properties of wheat flour dough [76] by reinforcing intermolecular disulfide bonds in gluten proteins and forming stronger protein-starch networks. Longer treatment times did not yield additional benefits, while the increased flour hydration due to plasma treatment also played a role in enhancing dough properties. These findings have implications for optimizing dough preparation and enhancing the quality of baked products.

Other studies [93,94] introduce an innovative approach for grafting fruit trees, specifically pear and cherry trees, utilizing DBD technology. The research demonstrates that applying plasma treatment to the scion and rootstock cuts of the grafting process yields several advantageous outcomes. One of the key findings is that plasma-treated grafting leads to the creation of a more robust and qualitatively improved grafting area. This enhancement contributes to a faster accumulation of plant biomass. Furthermore, the impedance spectroscopy [113,114] revealed that plasma treatment contributes to the formation of a more advanced vascular system within the grafting area.

### 4.2. Atmospheric Pressure Plasma Jets and Plasma Torch

The study [96] identified the optimal APPJ parameters that promoted growth stimulation. The changes observed in the Mung bean (*Vigna radiata*) seed coat and increased hydrophilicity likely played a vital role in the improved water uptake by the seeds, contributing to their enhanced growth—for the onion seeds, such optimization was made in [97].

Treatment can significantly promote the propagation of protocorms in *Cymbidium tracyanum* L. Castle orchids [98]. The observed improvements in size, bud count, and weight, along with the cellular changes, were observed under the microscope.

Short treatment duration of APPJ fed by microwave [99] demonstrated effective inactivation of *A. niger*, a mold fungus, while its impact on the gram-positive bacteria *B. subtilis* was less significant, regardless of the treated object (berries). Extending the treatment time resulted in the elimination of mold fungi from the samples, but a complete antibacterial effect was not achieved. The chemical properties of the objects treated with plasma were either similar to or even better than intact samples, indicating that the treatment did not negatively impact the chemical quality of the spices. However, there was a noticeable change in the color of the samples subjected to plasma treatment, which may affect their visual appeal.

A positive impact on the antioxidant activity of herbal extracts is described in [95]. The structural changes in plant materials resulting from the processing process played a crucial role in extracting more biologically active compounds, particularly polyphenols. Specifically, the content of flavonoids and anthocyanins increased. However, the composition of volatile compounds in the extracts was reduced due to plasma treatment, leading to a less intense smell of the extracts. Furthermore, the use of plasma resulted in a decrease in the total number of aerobic bacteria in the extract solutions. Additionally, the color of the extracts changed, and there was an increase in the pH of the solutions.

### 4.3. Inductively Coupled Radio Frequency Plasma

Inductively coupled radio frequency plasma source is implemented in a series of papers [101,102,103,104]. Thus, treatment significantly stimulated the accumulation of steviol glycosides, particularly stevioside, in the *Stevia rebaudiana* seeds. The rebaudioside A/stevioside ratio decreased, indicating a shift in the glycoside composition towards more stevioside. However, the treatment had unfavorable effects on total phenols, flavonoids, and antioxidant activity. In the case of the wheat seeds, only long-term direct treatment-induced changes in the morphology of the seed pericarp, leading to an increase in their roughness. This was accompanied by a decrease in the contact angle, indicating increased hydrophilicity and, subsequently, an increase in the water absorption of the seeds. The changes in the seed pericarp likely contributed to the observed increase in water uptake. However, the long-term plasma treatment had some negative effects on the germination of seedlings. It slowed down the germination process, suggesting that the treatment might have affected the early stages of seedling development. The plasma treatment also reduced the activity of α-amylase. The decrease in its activity could have implications for the availability of nutrients during germination. Furthermore, the root system of seedlings was affected by the plasma treatment. It is possible that the changes in seed pericarp and altered water absorption might have influenced root development. These findings indicate the need for careful consideration and optimization of plasma treatment parameters for seeds to avoid undesirable effects on seedling growth and development.

The study on high-intensity plasma treatment of bean seeds [106] demonstrated positive effects such as reduced fungal infection, increased wettability, changes in the seed surface, a drastic reduction in the contact angle of water, partial hydrophobic recovery over time, and an increase in root length. In corn seeds [105], the combination of the fungicide and plasma coating demonstrates a synergistic effect, providing a more comprehensive and robust defense mechanism against Fusarium contamination.

The red clover seeds [107] had a color-dependent response to plasma treatment. The differences observed in the germination kinetics, phytohormone content, and root nodule formation suggest that seed color may play a role in modulating the effects of plasma treatment on seed germination and early seedling development.

Different winter wheat varieties demonstrate the variability in responses to plasma seed treatment [108]. It also underscores the potential benefits of indirect treatment using afterglow plasma mode for enhancing the germination and early growth characteristics of the wheat. Another study [109] highlights the significant impact of electrical discharge on hormonal and metabolic regulation during wheat seedling germination.

### 4.4. Corona and Glow Discharges

Corona discharge [100] treatment was observed to induce structural changes in lentil seeds, leading to improved germination and growth rates. Notably, using cold plasma-activated wastewater from poultry farms was found to be the most effective medium for stimulating plant growth when compared to drinking rainwater. Furthermore, the treatment resulted in a decrease in the content of bacteria, specifically Escherichia coli and Salmonella typhimurium, in the lentil structures.

Low-frequency glow discharge plasma treatment [110], especially with an Ar + O_2_ gas mixture, also positively impacts seed germination, seedling growth, antioxidant enzyme activities, and grain quality in wheat. Ref [111] revealed significant changes in various biological characteristics and yield components. Specifically, the germination index, plant height, bushiness, growth, and number of grains showed notable alterations after the glow discharge plasma treatment.

Direct plasma treatment [112], in turn, induced noticeable morphological changes on the surface of seeds. These changes may include alterations in the seed coat or other structural modifications that could be related to improved water absorption, nutrient uptake, or protection against environmental stresses. Such morphological changes could be key contributors to the observed increase in yields and long-term treatment effects.

## 5. Plasma-Activated Water Treatment Results

A brief review of the indirect plasma treatment results that is, with PAW or PTS, in recent years is presented in Table 2.

### 5.1. Dielectric Barrier Discharge

Ref [115] investigated the processing of fresh-cut potatoes using PAW prepared by decreasing discharge frequency. Two different frequencies, 200 Hz and 10 kHz, were used for the plasma treatment, and their effects were compared. The study suggests that the plasma-activated water prepared with a lower discharge frequency of 200 Hz is more effective in disinfecting fresh-cut potatoes and exhibits better antioxidant properties. Additionally, both 200 Hz and 10 kHz PAW can effectively inactivate enzymes that cause browning in the potatoes, contributing to their extended shelf life and improved visual quality during storage.

PAW demonstrates, especially when enriched with magnesium and zinc ions, a positive influence on the growth and development of Chinese cabbage plants [116]. The treatment results in faster seed germination, enhanced seedling growth, increased chlorophyll and protein content, and favorable gene expression patterns, which collectively lead to healthier and more robust plants.

The beneficial effect on lettuce seed germination and subsequent seedling growth is described in [121]. Short exposure times appear to be more effective for promoting germination and growth, while longer exposure times may not yield further benefits. The increased chlorophyll content and positive morphological changes in the seeds demonstrate the potential of PAW as a tool to enhance the performance of lettuce crops.

Foliar application of PAW on maize plants [123] can lead to changes in chlorophyll content, photosynthetic efficiency, and nutrient composition, which may influence the plant’s growth and physiological processes.

Treatment of soybean seeds can promote faster germination and growth [125]. Moreover, the presence of ZnO nanoparticles can reduce the uptake of heavy metals by soybean plants, which may have implications for reducing heavy metal contamination in the soil and enhancing plant health and growth in contaminated environments.

The study demonstrates that treatment with PAW can effectively preserve strawberries [117], maintaining their quality and preventing spoilage. The PAW-treated strawberries did not undergo any negative changes in their sensory attributes, and the best quality indicators were observed on the fourth day after processing. The use of PAW in treating button mushrooms [118] proved to be beneficial in reducing browning and preserving organoleptic properties.

The influence on plant calcium signaling was investigated in [119], demonstrating that the chemical composition of PAW and the frequency of treatments can significantly affect the intracellular Ca^2+^ signals in plants.

PAW treatment positively influenced the photo-dependent dormancy mechanisms in Nicotiana tabacum seeds [120]. It enhanced testa and endosperm rupture percentages, increased the activity of the gibberellin-3-oxidase gene, and upregulated the expansin-A4 gene, all of which contributed to improved seed germination and growth.

It was found in [122] that PAW treatments had an impact on root hair density in Arabidopsis thaliana plants, and this modulation was associated with changes in the expression of certain root developmental genes. The COBL9, XTH9, and XTH17 genes were identified as potential target genes that were influenced by the PAW treatments. These genes play essential roles in root development and root hair formation. The study observed that PAW prepared with a short duration of plasma treatment led to an up-regulation of these target genes, indicating enhanced root hair development. On the other hand, the long-exposed PAW suppressed root development.

Studying the physical and physical-chemical properties of the soil under the PAW influence [124] indicates that the application of PAW had minimal impacts on soil, except for an improvement in water retention. This suggests that PAW may be a potentially useful tool for enhancing soil water retention and moisture availability for plants without causing significant changes to other soil characteristics.

### 5.2. Atmospheric Pressure Plasma Jets and Plasma Torch

PAW showed itself as an effective tool in degrading pesticide residues on kumquat fruits [126]. Additionally, it helps preserve soluble solids, increases titratable acidity, and does not affect fruit color change.

Ref [128] shows that treatment can affect the antioxidant contents in water spinach, particularly in the presence of heavy metals in the soil. PAW treatment has the potential to increase the total content of phenols and flavonoids in water spinach plants when certain heavy metals are present in the soil. However, the effects of the treatment on antioxidant contents can vary depending on the specific heavy metal present in the soil, indicating that the interactions between PAW and heavy metals play a crucial role in modulating the antioxidant response. Ref [140] shows that treatment can be beneficial in reducing the accumulation of Cd in water spinach, but its effect on Pb uptake and overall biomass may not be as significant.

The use of PAW as a nitrate source for hydroponically grown green oak lettuces showed promising results in terms of growth and nutritional quality [129]. It did not negatively affect lettuce growth and yield and may lead to a reduction in nitrate residues and an increase in essential amino acids in the lettuce. A positive effect on the growth of lettuce plants was also observed in [131]. However, this effect diminished over time, and there were no substantial differences in the root system between the PAW-treated and control plants. Additionally, PAW treatment resulted in higher dry matter content in the lettuce plants.

Pre-treated pepper seeds [132] and pea seeds [136] had a significant positive impact on the growth and yield of plants.

The combined effects of glow discharge plasma seed treatment and foliar application of PAW on paddy plant growth and yield were studied [139]—the air plasma seed treatment involved exposing rice seeds to low-pressure plasma for 90 s. The combination of plasma seed treatment and PAW application had a significant impact on plant growth parameters. The treated plants exhibited improved growth, which can be attributed to the synergistic effects of both treatments. Additionally, the study found that the combined treatment enhanced plant defense mechanisms by increasing enzymatic activity. This indicates that the plants had a more robust defense system against potential pathogens and stressors. The concentration of total soluble protein and sugar in rice grains was increased after the combined treatment, which suggests an improvement in the nutritional quality of the rice.

Ref. [142] aimed to investigate the effects of plasma-assisted nitrogen fixation on corn plant growth and development. The experiments involved using deionized water and water enriched with magnesium (Mg), aluminum (Al), or zinc (Zn) cations. These metal cations neutralized the PAW and enhanced the reduction of nitrogen to ammonia, with additional hydrogen generated from the reaction between the acid produced by the plasma and the metal ions. The results showed that corn seeds watered with PAW had a faster germination rate and more efficient growth, especially in the presence of metal ions. Moreover, high nitrogen concentrations in the growth medium increased the chlorophyll and protein content in the green parts of the corn plants. This increase in chlorophyll and protein levels led to the formation of intensely green leaves, indicating enhanced photosynthesis and overall plant health. The paper comprehensively discussed the chemical processes in water and the role of the metal’s oxidation levels. Metals immersed in water undergo oxidation, resulting in the generation of metal ions. These metal ions subsequently convert nitrite (NO_2_^−^) and nitrate (NO_3_^−^) species into their corresponding metal nitrates and nitrites, respectively. Concurrently, the electrons from the metal species lead to the reduction of hydrogen ions (H^+^) to hydrogen (H), thereby inducing a substantial increase in the solution’s pH level. Hydrogen facilitates the reduction of nitrogen to yield ammonia (NH_3_). As a consequence, the rates of ammonia synthesis and the degree of pH elevation within various activated waters follow the sequence: Zn-PAW < Al-PAW < Mg-PAW. At restricted concentrations, metal ions such as magnesium ions (Mg^2+^), aluminum ions (Al^3+^), and zinc ions (Zn^2+^) serve advantageous functions in plant physiology.

The impact on cytosolic calcium levels in Arabidopsis thaliana was investigated in [143]. The results showed that exposure to PAW induced a rapid and sustained increase in cytosolic calcium concentration in the plant cells. The specific features of this response were found to be influenced by various factors, including the operating conditions of the plasma torches used to generate PAW, the duration of plasma exposure to water, the amount of PAW applied to the plants, and the temperature and time of PAW storage. When the individual components of PAW (nitrates, nitrites, and hydrogen peroxide) were administered separately at the same concentrations as PAW, no significant changes in cytosolic calcium dynamics were observed. This suggests that the combination of these components in PAW plays a unique role in triggering the calcium response.

PAW treatment indicates strong antibacterial properties [127] on suspension cells and *P. fluorescence* biofilms and outperforms analog mixtures of traceable ingredients commonly used for their antimicrobial effects. Ref [130] demonstrated that PAW treatment mediates environmentally transmitted pathogenic bacterial inactivation through intracellular nitrosative stress, leading to a decrease in bacterial viability and causing morphological changes in the bacterial cells. Ref [133] also suggests that PAWs have strong antibacterial properties and low cytotoxicity, making them promising candidates for various biomedical and disinfection applications, especially when the pH is carefully controlled.

Plasma-activated acidic electrolyzed water (PA-AEW) was successfully used [134] as a food disinfectant for bacterial suspension and biofilm. The results showed that PA-AEW had a significant bactericidal effect against B. subtilis, a common bacterium used as a model microorganism in research. The sterilization time for PA-AEW was only 10 s, and during this short exposure, it demonstrated a satisfactory bactericidal effect. The effectiveness of PA-AEW in killing B. subtilis was found to be higher compared to other treatments, including PAW and acidic electrolyzed water.

The application of PAW against *Escherichia coli*, *Colletotrichum gloeosporioides*, and the decontamination of pesticide residues on chili (*Capsicum annuum* L.) was investigated in [135]. The results showed that treatment for 30 min and 60 min effectively degraded carbendazim and chlorpyrifos. In chili, the levels of carbendazim and chlorpyrifos were significantly reduced. Moreover, the treatment demonstrated strong antimicrobial activity. The populations of *Escherichia coli* were reduced by 1.18 Log CFU/mL in suspension and by 2.8 Log CFU/g in chili after 60 min of treatment. Additionally, the treatment achieved 100% inhibition of fungal spore germination, indicating its potential as an effective method to control fungal pathogens such as *Colletotrichum gloeosporioides*.

The investigation focused on nitrate capture in PAW, and its antifungal effect on *Cryptococcus pseudolongus* cells was observed in [137]. The study found that enriching PAW with magnesium ions (PAW-Mg^2+^) resulted in the control of free nitrate through the formation of nitrate salts by magnesium ions. Both PAW and PAW-Mg^2+^ exhibited antifungal activity against *C. pseudolongus*. However, the efficacy of PAW-Mg^2+^ was found to be lower compared to PAW alone. This suggests that the antifungal effect of PAW can be influenced and controlled by the presence of captured nitrate.

Ref [138] described the effectiveness of treatment in inactivating *Candida albicans* and spoilage fungi from lemon (*Citrus limon*). The results showed that PAW was highly effective in reducing the population of *C. albicans*. Even with short treatment times, PAW achieved a reduction of more than 6 log10 CFU/mL of *C. albicans*. PAW treatment caused damage to the membrane of *C. albicans* cells. This damage led to the leakage of cellular contents and ultimately resulted in the death of the cells. Furthermore, the study demonstrated the long-term fungicidal efficacy of PAW against *C. albicans* and the spoilage fungi found in lemon. This suggests that PAW treatment can provide sustained protection against these fungi, making it a promising option for food preservation and hygiene applications.

Ref [141] identified two critical processes that led to the death of *A. brasiliensis* spores after PAW treatment. First, the structural damage to the spore cell wall allowed the ROS and RNS in the PAW to penetrate and attack the spore’s internal components. Second, the genomic DNA of the spores was degraded by the ROS and RNS present in the PAW, further contributing to cell death.

The research findings [144,145,168] indicate that when plasma-activated water is diluted to concentrations of 0.5–1.0% in distilled water, it leads to several positive effects on seed germination and plant growth for various crops, including cotton, wheat, and strawberries. Specified dilution levels have been shown to enhance germination energy. This likely results in quicker and more vigorous sprouting of seeds, which is crucial for establishing healthy and productive plant populations. The treatment provides protection against fusariosis, a fungal disease caused by Fusarium species, and hyperthermia (heat stress) in seeds. PAW demonstrates advantages over conventional seed germination stimulants such as Dalbron and Bakhor. While these traditional stimulants may have been effective, the use of plasma-activated water offers additional benefits that might include enhanced disease resistance, stress tolerance, and overall growth stimulation.

### 5.3. Corona Discharge

Longer PAW treatment times positively preserved certain vitamins and polyphenolic compounds in arugula leaves [146]. Specifically, ascorbic acid, riboflavin, nicotinic acid, and nicotinamide were better preserved in leaves treated with PAW for a longer duration (20 min). Furthermore, the longer treatment with PAW increased the content of vitamins B2 (riboflavin) and B3 (nicotinic acid), as well as some individual polyphenols. This suggests that PAW treatment can enhance the nutritional content of arugula leaves. Interestingly, PAW treatment led to a significant decrease in antioxidant activity. This could be due to the degradation or modification of certain antioxidant compounds during the treatment process. Additionally, the study found a decrease in catalase activity in arugula leaves treated with PAW. Catalase is an enzyme involved in antioxidant defense, and its decrease may be related to the observed reduction in antioxidant activity. PAW treatment also has potential benefits for fresh-cut vegetables such as arugula by preserving their nutritional properties [148], increasing total phenols and glucosinolates, and improving the redox status without inducing cytotoxicity.

PAW generated with corona discharge, as with others, had a positive impact on wheat grain germination and seedling growth [147]. PAW exposure increased the germination rate, shoot length, and the fresh and dry mass of shoots. In addition, PAW also demonstrated effective decontamination properties against bacterial and yeast pathogens on artificially infected wheat grains. Wheat grains infected with Escherichia coli were effectively decontaminated after just 1 h of exposure to PAW. On the other hand, decontamination of grains infected with *Saccharomyces cerevisiae* required soaking the grains in PAW for 24 h.

### 5.4. Gliding Arc Discharge

The effect of PAW on maize seed germination and growth was investigated in [149]. It was shown that treatment of corn seeds with 15 min exposed PAW resulted in significant changes to the seeds, while a 5 min exposed PAW did not cause notable alterations.

For Beta vulgaris seeds [150], the PAW treatments resulted in a higher germination rate than the treatment with sodium hypochlorite (NaClO). Although the average seedling length was slightly shorter than the NaClO treatment, it was still promising for seedling growth. However, in the case of Daucus carota seeds treated with PAW, the germination and seedling length were lower compared to using a sodium hypochlorite solution but still significantly higher than in the control group. Moreover, PAW treatment affected the composition of fungal species present in the seeds of Beta vulgaris and Daucus carota. Different fungal species responded differently to PAW treatment, and for some types of fungi, the effect was not significant.

Basil plants grown using the plasma-activated nutrient solution (PANS) exhibited significant growth improvements compared to plants treated with regular nutrient solutions [151]. Specifically, the basil plants grown with PANS were taller and had a significantly higher dry weight. The study also evaluated the organoleptic characteristics. It was observed that these plants had significantly greener leaves compared to the control group. Furthermore, the leaves of basil plants grown with PANS showed higher levels of linalool and methyleugenol, which are compounds responsible for the plant’s floral and spicy aroma. This indicates that the plasma treatment positively influenced the aromatic properties of the basil. Another significant finding of the study was related to the reduction of algae concentration in the hydroponic medium. Repeated treatment of basil plants with PANS helped mitigate algae growth, contributing to a cleaner and more controlled hydroponic environment.

Seeds treated with PAW-5 (5-min plasma treatment) and PAW-10 (10-min plasma treatment) exhibited significant enhancements in various germination parameters [152]. Specifically, they had a higher percentage of germination and germination uniformity and increased average daily germination and germination values compared to the control group. The study evaluated the effect of PAW irrigation on different types of seeds, including barley, mustard, and rayo seeds. Irrigation with PAW-10 resulted in a significant increase in germination for all three types of seeds. Moreover, seeds watered with PAW showed higher water uptake, longer roots, and longer shoots compared to the control group. This indicates that PAW irrigation positively affected seedling development and growth. Additionally, vigor analysis demonstrated that seeds irrigated with PAW exhibited stronger growth characteristics.

### 5.5. Spark and Glow Discharges

The research [153] evaluated different rice starch-phenolic complexes and assessed the contributions of different factors, including the type of phenolic compound, PAW, and ultrasonication. The addition of gallic acid to starch molecules resulted in an increase in the activity of removing 1,1-diphenyl-2-picrylhydrazyl (a common method to measure antioxidant activity). However, it was observed that the complexes with gallic acid had lower values of the complexation index and the content of resistant starch compared to the complexes with crude Mon-pu extract.

The PAW effect on the chemical compounds present in the leaves of *Eruca sativa* (rocket salad) was investigated in [154]. Different PAW was used to study their impact on the volatile organic compounds (VOCs) in the leaves. A total of 52 VOCs from various chemical classes were detected and quantified using gas chromatography-mass spectrometry. PAW treatment can induce chemical modifications in the VOCs. The specific changes in compound content varied with different exposure times, and some compounds showed significant increases while others decreased. The decontamination effect was achieved after washing the rocket leaves with PAW [155]. This did not lead to significant changes in the quality and nutritional parameters. There were only slight changes in color and the content of bioactive compounds. Moreover, the antibacterial effect of PAW treatment was found to be more pronounced compared to the use of hypochlorite, which is a commonly used disinfectant in the food industry.

A comparison of the effect with a chemically equivalent solution of hydrogen peroxide (H_2_O_2_) + nitrates (NO_3_^−^) on the growth and physiological parameters of lettuce plants is presented in [159]. When lettuce plants were irrigated with PAW and the chemically equivalent H_2_O_2_ + NO_3_^−^ solution, they showed similar dry weight of above-ground parts and roots. However, there were notable differences in certain physiological parameters. The lettuce plants irrigated with PAW had a higher content of photosynthetic pigments (chlorophyll a + b) and exhibited a higher photosynthetic rate, and the activity of antioxidant enzymes, such as superoxide dismutase, was lower.

Ref [156] investigated that PAW treatment can positively influence the early growth stages of maize seedlings under arsenic stress. The study also highlights the complex interplay between PAW treatment and arsenic stress on various physiological and biochemical aspects of maize plants.

Application of PAW to pea seedlings [157] stimulated amylase activity, indicating enhanced starch breakdown without negatively affecting seed germination, total protein concentration, and protease activity. Furthermore, PAW had almost no oxidative stress on pea seedlings. In pea seedlings, PAW treatment resulted in a more rapid transition from anaerobic to aerobic metabolism, as evidenced by the inhibition of alcohol dehydrogenase activity. Additionally, the presence of reactive oxygen/nitrogen species in PAW did not affect DNA integrity in pea seedlings. However, the response of barley seedlings was different. They showed high levels of DNA damage, along with a decrease in the length of roots and shoots and a decrease in amylase activity.

Pea plants grown from seeds pre-treated with PAW showed enhanced growth compared to untreated seeds [112]. This improvement was particularly noticeable in the number of seeds per pod and the total number of seeds per plant. Morphological changes on the seed surfaces may include alterations in the surface structure, texture, or microstructure of the seeds. Such changes could potentially contribute to the improved growth and yield observed in the treated plants.

Significant changes in the concentration of hydrogen peroxide (H_2_O_2_) and ROS were observed in the seeds, leaves, and roots of the black gram plants [161]. Specifically, an increase in the level of catalase enzyme was observed in the roots of plants grown from seeds treated with PAW for 3 and 6 min. This increase in catalase activity is consistent with the activation of the VmCAT gene, which encodes for the catalase enzyme in black gram.

The study [163] indicates that PAW has both positive and negative effects on *Triticum aestivum*. While it may promote the germination of wheat grains and the growth of roots and shoots at specific concentrations and pH levels, it also demonstrated cytogenotoxic potential, indicating possible harmful effects on cell division and genetic material.

The application of PAW and the biostimulator [160] decreased the incidence of fungal diseases on the lawn, positively influenced turf density, led to a thicker and more lush lawn, and positively affected overwintering.

A reaction-discharge system optimization to continuously produce a PAW with specific physicochemical and anti-phytopathogenic properties was performed in [158]. PAW was found to have bacteriostatic and bactericidal effects against two phytopathogenic strains, *Dickeya solani* IFB0099 and *Pectobacterium atrosepticum* IFB5103.

The scalable treatment of flowing organic liquids using ambient-air glow discharge for agricultural applications was a focus in [162]. The plasma treatment of the L-phenylalanine solution had a dual effect. On one hand, it stimulated hydroponic radish seedlings, resulting in a 40% increase in growth. On the other hand, it also demonstrated a bactericidal effect on *E. coli* (specifically, *E. coli* O1:K1:H7), leading to a reduction in bacterial load.

### 5.6. Underwater Discharge

Ref. [164] highlights the beneficial effects of spraying plasma-activated water on potato plants, leading to enhanced growth, biochemical activity, nutritional composition, and yield of the potato crop.

The series of papers [165,166,167,169] highlights the diverse applications of PTS in agriculture. PTS can be utilized as both a disinfectant and a nutrient solution for various agricultural practices. It can be used either in its freshly prepared form for disinfection purposes or diluted for irrigation, hydroponics, aeroponics, and mist generation. In the case of apple fruits, PTS has been shown to impact the calcium (Ca) content of the fruits when used for irrigation. This suggests that PTS application could have a positive effect on fruit quality and potentially extend the storage duration of the fruits. PTS has also been demonstrated to be effective in enhancing sorghum seed germination. Additionally, the treated plants displayed drought resistance. The use of the technique has been extended to field experiments conducted in a saline semi-desert environment. This implies that PTS could have applications in regions with difficult growing conditions, potentially helping plants adapt to challenging soil and environmental conditions.

### 5.7. H_2_O_2_ and NO_x−_ Generation Efficiency for Different Types of Plasma Sources

The use of PAW and PTS in agrobiological tasks and the study of plant physiology necessitates the accurate determination of the concentrations of ROS, RNS, and other bioactive compounds and nanoparticles that arise during plasma treatment. We analyzed dependencies on the energy input for hydrogen peroxide generation, as well as the concentrations of nitrite and nitrate ions in different types of water. The results have been categorized based on the type of plasma source, as previously described. The results at two different scales are shown in Figure 4 and Figure 5. It is important to note that the majority of investigated studies do not focus on the energy optimization of ROS and RNS generation. Furthermore, ROS and RNS production depends on a multitude of system parameters: interaction surface area with the liquid, the distance between electrodes and liquid, gas composition in the plasma–liquid interaction region, gas flow rate (if applicable), liquid flow rate (if applicable), electrode material (for immersed electrodes), and more. Therefore, these results serve to illustrate the most general trends.

The analysis, first, shows the need to standardize the description of plasma source characteristics, treatment methods, and ROS and RNS concentrations in PAW. Without this standardization, comparing experimental results and, more importantly, reproducing them using different plasma sources becomes challenging, significantly impeding the integration of plasma technologies into widespread practice.

Regarding PAW creation, all types of plasma sources demonstrate their viability. Each type has its advantages and drawbacks, which define their respective areas of application and demand.

DBD shows good energy efficiency when the goal is not to achieve ROS and RNS concentrations exceeding 5−6 mM and when large volumes of PAW are not required. To enhance productivity, increasing the plasma–liquid interaction surface area is necessary, transitioning from a planar electrode configuration to, for example, a coaxial one and implementing liquid flow.APPJ also appears attractive in terms of energy efficiency. Using gases to create a plasma jet offers broad possibilities in enriching PAW with target active compounds. For instance, it is feasible to achieve high RNS production while keeping the liquid free from extraneous impurities. On the other hand, using a gas injection system places additional demands on workspace organization and slightly raises treatment costs. For creating larger volumes of PAW, using a microwave plasma torch as the plasma source seems more efficient.Corona, gliding, and spark discharges might not initially show high ROS and RNS production, but they provide the opportunity to create “pure” PAW without side impurities, which is in demand, for example, in medicine or food processing. To improve the energy efficiency of PAW generation, increasing the plasma–liquid interaction area can be achieved through multi-spark (multi-electrode) systems. Gas injection into the discharge region is also described to enhance the production of target bioactive compounds.Underwater discharges allow for the generation of large volumes of PAW with high concentrations of ROS and RNS. These discharges can be employed for creating concentrates to replace chemical fertilizers in agrobiological tasks or for disinfection purposes. However, close attention must be paid to studying electrode erosion and electrolysis processes, which significantly contribute to the chemical composition of PAW. The presence of metal nanoparticles and dissolved metal compounds can either help or hinder the benefits of PAW use.

## 6. Conclusions

Analysis of scientific literature reveals that applying plasma in agriculture can contribute to improving numerous key indicators in crop production. This becomes particularly relevant due to the global demands of food production while ensuring environmental preservation. However, despite its considerable potential, plasma-based agriculture confronts a series of complex challenges that warrant comprehensive attention to facilitate its successful integration into mainstream agricultural practices.

One of the primary challenges lies in the development and selection of plasma sources suitable for agricultural applications. These sources must generate stable and controllable plasma states while being energy-efficient, cost-effective, and scalable for deployment across various farming scales.Achieving consistent and reproducible results with plasma treatments across different settings remains a challenge. Standardizing parameters such as discharge power, treatment duration, and gas composition is essential to ensure the reliability and comparability of outcomes.Clarifying the underlying physiological, biochemical, and genetic mechanisms of plasma–plant interactions is critical for maximizing desired effects. The intricate interactions between plasma and plants are not yet fully understood. The variability of plant species, growth stages, and environmental conditions can influence the outcomes of plasma treatment.Determining the optimal treatment conditions for diverse agricultural contexts is complex. Tailoring plasma treatments to address specific crop types, growth stages, and environmental factors requires a nuanced understanding of how these variables influence plasma effects.In a broad range of tasks, direct plasma treatment can be substituted with the use of PAW treatment. However, it is crucial to identify and control the ROS and RNS and the dissolved ions of metals and nanoparticles of metals and other elements present initially in the electrodes and vessels used for generating PAW.Achieving precise and reproducible biological effects from the use of PAW requires studying the dynamics of changes in the concentration of ROS and RNS over the period necessary for delivering PAW to the consumer. Research demonstrates that the concentration of several important bioactive compounds can change by an order of magnitude (or more) within several tens of minutes after the completion of plasma treatment.

Finally, evaluating the long-term effects of plasma treatments on soil health, plant growth, and ecosystem resilience is also important. Uncovering potential cumulative effects or alterations to natural systems is essential to ensure sustained benefits.

## Figures and Tables

**Figure 1 ijms-24-15093-f001:**
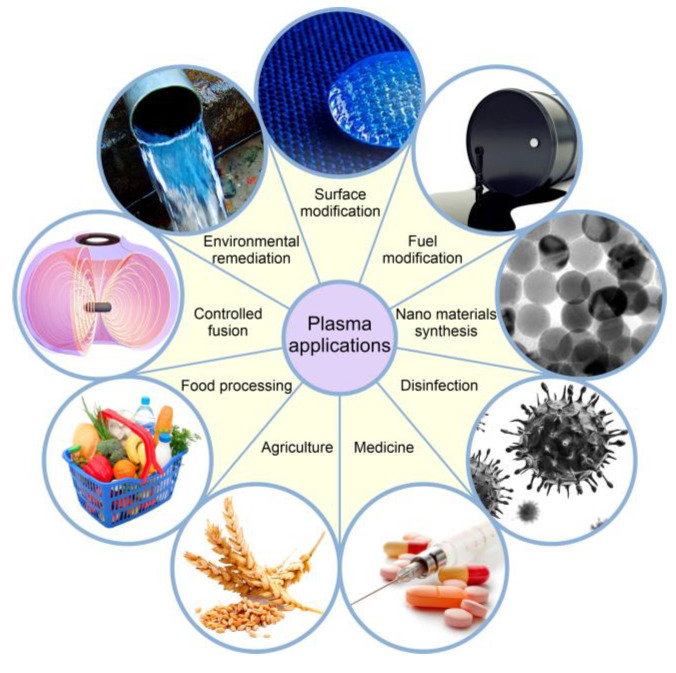
Application of plasma technologies.

**Figure 2 ijms-24-15093-f002:**
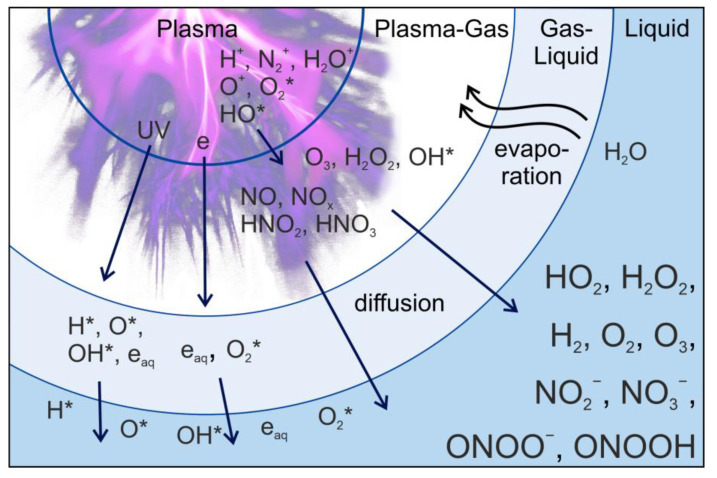
Simplified model of plasma–liquid interaction. * indicates radical.

**Figure 3 ijms-24-15093-f003:**
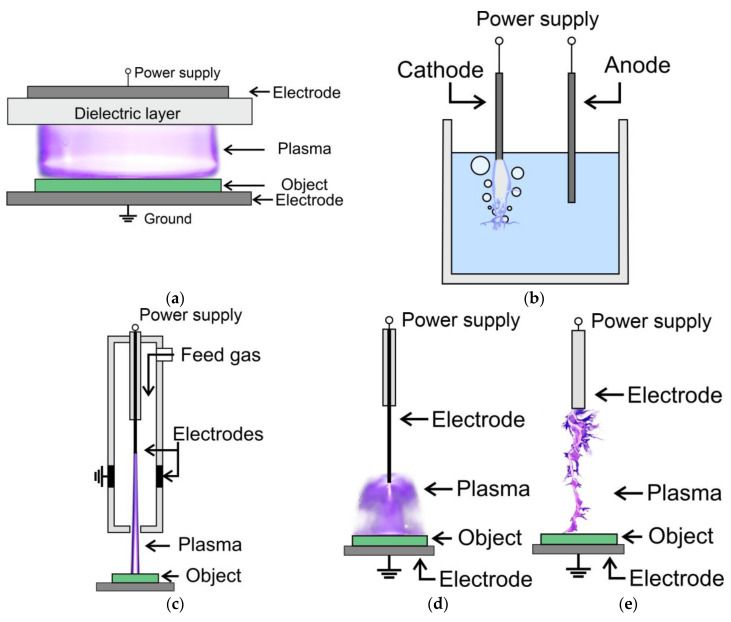
General configuration of primary types of plasma sources in agriculture and biology: (**a**) dielectric barrier discharge; (**b**) underwater discharge; (**c**) atmospheric pressure plasma jet; (**d**) corona discharge; (**e**) spark discharge.

**Figure 4 ijms-24-15093-f004:**
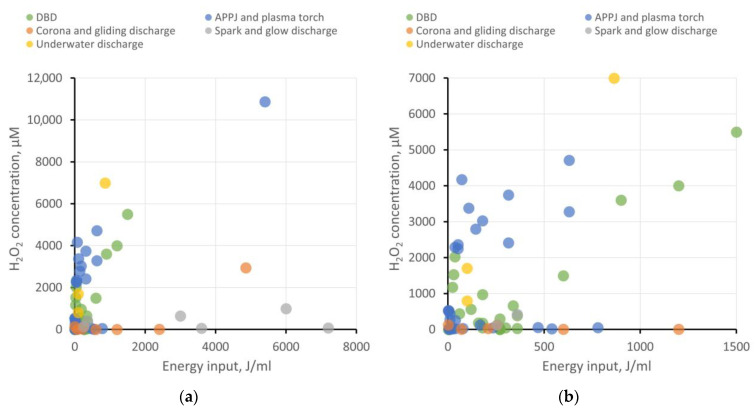
Production of hydrogen peroxide in water depending on energy input per milliliter of water for various types of plasma sources. For the convenience of analysis, the results are presented on two scales: (**a**) energy input up to 8000 J/mL and (**b**) energy input up to 1500 J/mL.

**Figure 5 ijms-24-15093-f005:**
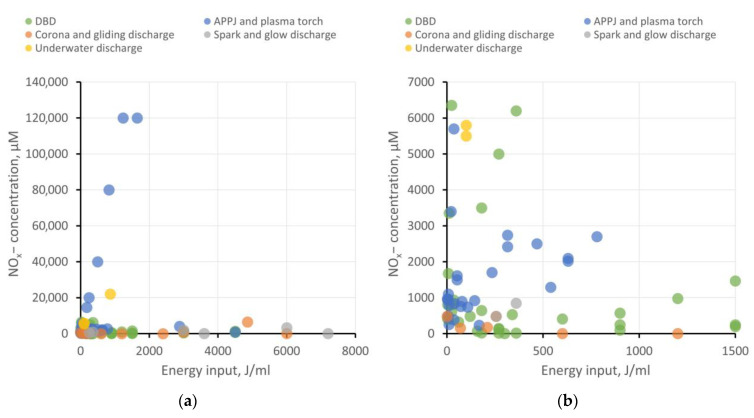
Production of NO_x_^−^ ions in water depending on energy input per milliliter of water for various types of plasma sources. For the convenience of analysis, the results are presented on two scales: (**a**) energy input up to 8000 J/mL and (**b**) energy input up to 1500 J/mL.

**Table 1 ijms-24-15093-t001:** Overview of direct plasma treatment results in recent years.

Plasma Source Parameters	Object	Key Results	Reference
DBD	Soybeans (*Glycine max* L.)	Increased protein extraction yield, water binding and oil absorption capacities, and improved emulsifying activity.	[67]
DBD	Spinach seeds	Increased water absorption and germination.	[68]
DBD	Sunflower (*Helianthus annuus* L.) seeds	Faster growth, taller seedlings, and increased yield in mature plants persisting even after storage for two weeks.	[69]
DBD	Lettuce (‘Kerlis’)	Increased yield, soluble solids, conductivity, and acidity. High ionization plasma raised nitrogen, phosphorus, and potassium content and increased leaf pigments.	[70]
DBD	Wheat (*Triticum aestivum* L. ‘Dacic’ and ‘Otilia’) seeds	Increased chlorophyll and flavonoid content	[71].
DBD	*Brassica oleracea* and *Lepidium sativum* seeds	Enhancing hydrophilicity. Increased stem and root length in seedlings.	[72]
DBD	Eggplant (*Solanum melongena* L.) seeds	Increased seed germination plant growth, antioxidant activity, soluble sugars, proteins, total phenols, and certain mineral concentrations (Ca, Cu, Fe, Mn, K), while Zn content decreased.	[73]
DBD	White Radish (*Raphanus sativus*)	Changed seed coat morphology, enhancing water permeability. Improved germination, chlorophyll content, antioxidant activity, and seedling phenols.	[74]
DBD	Mushrooms (*Flammulina velutipes*)	Maintained weight parameters and superoxide anion formation rate. Improved antioxidant activity, enzyme activity, malondialdehyde levels, vitamin C retention, and extended shelf life.	[75]
DBD	Wheat (*Triticum aestivum* L.) flour	Improved wheat flour dough’s viscoelastic properties by strengthening gluten protein-starch networks. Increased flour hydration.	[76]
DBD	*Xylella fastidiosa*	Complete inhibition of bacterial growth.	[77]
DBD	Wheat (*Triticum aestivum* L.) seeds		[78]
DBD	Sundried tomatoes (*Solanum lycopersicum* L.), *Aspergillus rugulovalvus*, *Aspergillus niger*	Reduced bacterial and fungal populations. Lycopene content increased.	[79]
DBD	*Arabidopsis thaliana* seeds	Improved seed germination.	[80]
DBD	Basil (*Ocimum basilicum* L. ‘Genovese Gigante’)	Increased leaf humidity, chlorophyll, carotenoids, antioxidant activity, flavonoids, and peroxidase activity. Reduced microbial load.	[81]
DBD	*Arabidopsis thaliana*, *Camelina sativa* seeds	*A. thaliana* benefits from longer exposure due to increased unsaturated fatty acids, while C. sativa’s optimal time is shorter due to reduced unsaturated fatty acids.	[82]
DBD	*Pectobacterium carotovorum*, *Pectobacterium atrosepticum*, *Dickeya solani*, inoculated either on mung bean seeds	Inactivated bacteria. A 2-min exposure stimulated seed germination and growth. A 4-min exposure hindered germination and growth.	[83]
DBD	Dried maize (*Zea mays* L.) ‘Ronaldinio’ grains	Enhanced seed viability and seedling growth through enzyme activity stimulation. Triggered heat shock proteins with minimal DNA damage. Surface hydrophilicity increased.	[84]
DBD	Buckwheat (*Fagopyrum esculentum* Moench) ’VB Vokiai’ and ’VB Nojai’	Decreased emergence but increased growth, biomass, and yield. Enhanced seed weight per plant.	[85]
DBD	Dried pea (*Pisum sativum* L.) ’Prophet‘ seeds	Enhanced seed germination. Increased peroxidase activity in cell walls and mechanical strength.	[86]
DBD	Dried barley (*Hordeum vulgare* L.) ’Maltz‘ grains	Improved germination and enzyme activity.	[87]
DBD	Common pine (*Pinus sylvestris* L.), black pine (*Pinus nigra* Arnold), mountain pine (*Pinus mugo* Turra) seeds	Short treatment duration improved germinated and growing. Long treatment duration had a retarding effect.	[88]
DBD	Dried pea (*Pisum sativum* L.) seeds	Improved germination, minimized DNA damage	[89]
DBD	Winter wheat (*Triticum aestivum* L.)	Improved surface wettability and germination.	[90]
DBD	Pea (*Pisum sativum* L.) ’Prophet‘ seeds	Treatment for 60 s improved seed germination by enhancing surface wettability and activating enzymes. Shorter treatments stimulated germination without DNA damage.	[91]
DBD	Raw barley (*Hordeum vulgare* L.) grains	Reduced deoxynivalenol mycotoxin concentration.	[92]
DBD	Cuts of rootstock and scion of pear *(Pyrus communis* L.)	Enhanced scion growth. Improved vascular system differentiation.	[93]
DBD	Cuts of rootstock and scion of cherry	Enhanced scion growth. Improved vascular system differentiation	[94]
APPJ	Extracts from 12 herbs: *Echinacea purpurea*, *Salvia officinalis*, *Urtica dioica*, *Polygonum aviculare*, *Vaccinium myrtillus*, *Taraxacum officinale*, *Hypericum perforatum*, *Achillea millefolium*, *Sanguisorba officinalis*, *Leonurus cardiaca*, *Ballota nigra*, *Andrographis paniculata*	Enhanced antioxidant activity in extracts by promoting polyphenol extraction, increasing flavonoids and anthocyanins, while reducing volatile compounds and altering aroma. Lowered aerobic bacteria. Induced color and pH shifts.	[95]
APPJ	Mung bean (*Vigna radiata*)	Improved seed germination and stem length. The contact angle decreased, aiding water uptake.	[96]
APPJ	Bulb onions (*Allium cepa* L.) seeds	Improved germination and vigor.	[97]
APPJ	Orchid(*Cymbidium tracyanum* L. Castle) protocorms	Improved size, bud count, fresh and dry weights. Disrupted cell walls, aiding bud elongation and dormancy release.	[98]
Microwave-driven plasma jet	Whole black pepper seeds, whole allspice berries, and whole juniper berries	Inactivated fungi but did not achieve complete antibacterial effects.	[99]
Corona discharge	Lentil seeds	Improved germination and growth. Reduced bacteria.	[100]
DBD	*Stevia rebaudiana*	Enhanced seed water absorption, germination, and plant yield. Improved nutritional content and potential shelf life extension.	[101]
Inductively coupled RF discharge	Grains of common buckwheat (*Fagopyrum esculentum Moench*) infected with the following fungi: *Alternaria alternata* (GB002), *Aspergillus flavus* (GB005), *Aspergillus niger* (GB006), *Cladosporium cladosporioides* (GB007), *Epicoccum nigrum* (GB009), *Fusarium fujikuroi* (GB011), *Fusarium graminearum* (GB012), *Fusarium oxysporum* (GB013), *Fusarium proliferatum* (GB014), *Fusarium sporotrichioides* (GB015)	Reduced contamination for most fungal taxa, with *Fusarium graminearum* being the most sensitive and *Fusarium fujikuroi* the most resistant.	[102]
Inductively coupled RF discharge	Winter wheat (*Triticum aestivum* L.) ’Ingenio‘ seeds	Increased roughness and lowered contact angle. Enhanced water absorption. Hindered seedling germination, α-amylase activity.	[103]
Inductively coupled RF discharge	Alfalfa (*Medicago sativa* L.) seeds	Enhanced seed surface hydrophilicity.	[104]
Inductively coupled RF discharge	*Fusarium graminearum* and *Fusarium proliferatum* contaminated maize (*Zea mays*) seedlings	Fungicide (prothioconazole) combined with plasma effectively reduces fungi contamination.	[105]
RF discharge	Common bean (*Phaseolus vulgaris* L.),	Reduced fungal infection. Increased seed wettability. Decreased hydrophobicity. Increased root length.	[106]
Inductively coupled RF discharge	Red clover (*Trifolium pratense* L.) ’Arimaiciai‘ seeds	Improved germination. Phytohormone levels varied, not directly correlating with germination. Increased root nodule numbers.	[107]
RF discharge	Winter wheat (*Triticum aestivum* L.) ’Apache‘ and ’Bezostaya 1‘ seeds	Increased vigor index, root system, seedling wet weight, and germination rate.	[108]
High voltage electrical discharge	Wheat (*Triticum aestivum* L.) ‘BC Opsesija’	Enhanced germination and growth by altering hormone and metabolite levels.	[109]
Glow discharge	Wheat (*Triticum aestivum* L.) seeds	Increased seed germination. Superoxide dismutase, catalase, and ascorbate peroxidase activity increased in shoots. Increased content of soluble sugars, proteins, iron, manganese, fat, and ash. Moisture content decreased.	[110]
Glow discharge	Wheat (*Triticum aestivum* L.) ’Shannong 12’	Improved seed germination index, plant height, bushiness, growth, and number of grains.	[111]
Transient spark discharge	Pea (*Pisum sativum* L.) ’Eso‘ seeds	Increased yields and sustained effect.	[112]

**Table 2 ijms-24-15093-t002:** Overview of the indirect plasma treatment results in recent years.

Plasma Source	Liquid	PTS Characteristics	Object	Key Results	References
DBD	Distilled water	pH: 3.42 ± 0.19 and 2.64 ± 0.05; ORV: 461.67 ± 18.18 and 547.33 ± 9.02 mV; Ec: 182.67 ± 16.8 and 883.33 ± 37.21 µs/cm; [O_3_]: 1.21 ± 0.24 and 6.05 ± 0.73 mg/L; [H_2_O_2_]: 181.67 ± 40.41 and 658.33 ± 28.87µM; [NO_3_^−^]: 4.03 ± 0.19 and 32.45 ± 5.43 mg/L; [NO_2_^−^]: 0.31 ± 0.04 and 0.46 ± 0.08 mg/L, When exposed to 10 kHz and 200 Hz, respectively	Fresh-cut potato	Disinfection. Antioxidant properties. Enzyme Inactivation.	[115]
DBD	Deionized water	pH: 7.17 ± 0.34, 8.04 ± 0.51, 3.97 ± 0.065, 6.807 ± 0.210; ORP: 176.5 ± 3.53, 194 ± 5.65, 301 ± 1.41, 241.5 ± 0.70 mV; TDS: 16.9 ± 1.27, 26.5 ± 1.31, 51.6 ± 1.44, 30.2 ± 1.74 ppm; Ec: 21 ± 2.83, 33.7 ± 1.62, 74 ± 2.12, 43.3 ± 2.99 µS/cm [NO_x_^−^]: 0.52 ± 0.04, 0.452 ± 0.2, 130.61 ± 1.39, 144.37 ± 1.48 µM; [H_2_O_2_]: 0.417 ± 0.01, 0.447 ± 0.03, 4.5 ± 0.26, 3.875 ± 0.441 µM for deionized water with Met^+^, deionized water with Met^+^, PAW without Met^+^, PAW with Met^+^, respectively	Pak Choi seeds (*Brassica campestris* L.)	Faster germination. Increased seedling length. Higher chlorophyll and protein content. Positive gene expression.	[116]
DBD	Deionized water	[NO_3_^−^]: ~25, 35, 60, 90, 110 mg/L [NO_2_^−^]: ~0.15, ~0.4, 0.5, 0.82, 0.9 mg/L [H_2_O_2_]: <1, ~1.5, 3.6, 4.0, 5.5 mM [O_2_^−^]: ~26, 34, 22, 7, 5 mM for 1, 2, 3, 4, 5 min. DBD treatment, respectively	Strawberry	Longer shelf life, reduced spoilage. No taste/texture change. Best quality after 4 days of PAW treatment.	[117]
DBD	Pure water	pH: 5.16 ± 0.03	Button mushrooms (*Agaricus bisporus*)	Decreased champignon browning, inhibited enzymes, and maintained organoleptic quality for preservation.	[118]
DBD	Deionized water	[NO_3_^−^]: ~0.1, 0.2, 0.4, 0.7, 1.9 mM for 3, 5, 10, 15, 30 min. 12 kHz plasma treatment, respectively [NO_3_^−^]: ~0.25, 0.25, 1.2, 1.3, 3.3 mM For 3, 5, 10, 15, 30 min. 20 kHz plasma treatment, respectively	*Arabidopsis thaliana* seedlings	Low DBD-PAW doses influenced intracellular Ca^2+^ signals.	[119]
DBD	Distilled water	[H_2_O_2_]: 0, 0, 180.4 ± 7.2, 294.9 ± 18.4, 387.7 ± 24.5 μM [·OH]: 16.7 ± 3.3, 25.5 ± 2.9, 55 ± 4.0, 80.0 ± 2.9, 0 μM [NO_2_^−^]: 0, 0, 0, 0, 0 μM [NO_3_^−^]: 0, 0, 22.7 ± 3.6, 18.1 ± 3.8, 15.7 ± 1.9 μM With He/O_2_ plasma activation times of 10, 15, 30, 45, and 60 min, respectively [H_2_O_2_]: 0, 0, 47.8 ± 3.3, 33.3 ± 2.0, 30.2 ± 0.2 μM [·OH]: 14.5 ± 3.3, 23.1 ± 2.3, 42.7 ± 5.2, 54.3 ± 4.7, 0 μM [NO_2_^−^]: 0, 0, 56.5 ± 2.5, 47.3 ± 1.7, 35.4 ± 4.0 μM [NO_3_^−^]: 0, 0, 3420.7 ± 103.5, 4948.6 ± 74.5, 6191.1 ± 101.2 μM With air plasma activation times of 10, 15, 30, 45, and 60 min, respectively	*Nicotiana tabacum* ‘Havana 425’ seeds	Air and He/O_2_ PAW enhanced testa and endosperm rupture in low fluence conditions. Increased GA3ox2 and EXPA4 activity.	[120]
DBD	Distilled water	pH: ~5.8, 5.45, 4.8, 4.85, 4.8, 4.8 Ec: ~10, 13, 14, 15, 18, 17 μS/cm [H_2_O_2_]: ~15, 19, 33, 40, 52, 69 mg/L [NO_2_^−^]: <5 mg/L [NO_3_^−^]: ~20, 30, 40, 38, 50, 58 mg/L For 5, 10, 15, 20, 25, and 30 min plasma treatment, respectively	Lettuce (*Lactuca sativa* L.) seeds	Boosted lettuce seed germination with positive effects on seedling growth and chlorophyll content.	[121]
DBD	Deionized water	pH: 3.62 ± 0.02, 3.34 ± 0.03, 2.94 ± 0.08, 2.62 ± 0.07, 2.37 ± 0.04 Ec: 118.10 ± 2.26, 218.50 ± 9.64, 460.33 ± 15.25, 972.93 ± 32.41, 1847.00 ± 70.19 µS/cm [H_2_O_2_]: 0.09 ± 0.01, 0.14 ± 0.01, 0.27 ± 0.02, 0.88 ± 0.04, 1.31 ± 0.04 mg/L [NO_2_^−^]: 1.09 ± 0.11, 1.24 ± 0.12, 1.85 ± 0.07, 3.68 ± 0.12, 5.17 ± 0.16 mg/L [NO_3_^−^]: 25.29 ± 2.88, 49.05 ± 2.61, 102.67 ± 6.30, 204.87 ± 8.74, 389.08 ± 12.24 mg/L For 5, 7, 12, 19, and 40 min plasma treatment, respectively	*Arabidopsis thaliana* L.	Affected root hair density via gene regulation (COBL9, XTH9, XTH17).	[122]
DBD	Distilled water	[H_2_O_2_]: 0.7 ± 0.2 mg/L [NO_2_^−^]: 1.071 ± 0.005 mg/L [NO_3_^−^]: 24.7 ± 2.3 mg/L	Maize *(Zea mays* L. ‘SY ORPHEUS’*)*	Reduced leaf chlorophyll, changed fluorescence parameters, and increased nitrogen content.	[123]
DBD	Distilled water	pH: 6.7 Ec: 34 µS [H_2_O_2_]: 1.4 ± 0.4 mg/L [NO_2_^−^]: 0.753 ± 0.009 mg/L [NO_3_^−^]: 20.4 ± 1.8 mg/L	Soil	Minimal effects on soil, with slight changes in evaporation, pH, and water absorption. Higher PAW doses slowed tap water absorption but increased water retention.	[124]
DBD	Deionized water	pH: 4.3 [NO_3_^−^]: 25.7 mg/L [NO_2_^−^]: 16.4 mg/L [H_2_O_2_]: 2–5 mg/L	*Xylella fastidiosa*	Deactivation of *Xylella fastidiosa* cells.	[77]
DBD	Water	[H_2_O_2_]: 1,1,2,2,10,100, 100, >100, >100 ppm for 30 kV, 3 min., 30 kV, 5 min., 30 kV, 7 min., 50 kV, 3 min., 50 kV, 5 min., 50 kV, 7 min., 70 kV, 3 min., 70 kV, 5 min., 70 kV, 7 min. plasma treatment, respectively.	Soybeans (*Glycine max*)	Faster germination and growth. ZnO nanoparticles reduced heavy metal uptake in plants.	[125]
APPJ	Deionized water	pH: ~4.5, 3.5, 3 Ec: ~50, 270, 590 µS/cm For 10, 15, and 20 kV treatment, respectively	*Cuimi kumquat*	Reduced pesticide residues, preserved soluble solids, increased acidity, and maintained fruit color.	[126]
Microwave-driven plasma torch	Deionized water	[NO_3_^−^]: 72.3 mg/L [NO_2_^−^]: 1600.7 mg/L [H_2_O_2_]: 717.3 mg/L	*Pseudomonas fluorescence* suspended cells and *P. fluorescence* biofilms	Strong antibacterial effects. Effective against suspension cells and *P. fluorescence* biofilms.	[127]
APPJ	Reverse osmosis water	[NO_3_^−^]: 42.7 ± 0.70 mg/L [NO_2_^−^]: 14.7 ± 0.58 mg/L pH: 3.17 ± 0.06 Ec: 311.7 ± 12.01 µS/cm ORP: 554 ± 2.65 mV	Water spinach *(Ipomoea aquatica)* seeds	Phenols increase with PAW, more with Cd, and less with Pb. Flavonoids rise with PAW and Cd, with no change with Pb. Heavy metals impact PAW effects.	[128]
Pinhole plasma jet	Tap water	pH: 5.5–6 Ec: 1.5 µS/cm [NO_3_^−^]: 883.59 mg/L [NO_2_^−^]: 31.56 mg/L [H_2_O_2_]: 102.99 mg/L	Green oak lettuce (*Lactuca sativa* L.)	Growth parameters are mostly unaffected. Leaf area and greenness differed. Yields similar to commercial nitrate. Lower nitrate residues with PAW treatment. Plasma nitrate is converted to amino acids at higher concentrations than normal nitrate.	[129]
Microwave-driven plasma torch.	Deionized water	pH: ~4 ORP: ~502 mV Ec: ~1367 µS/cm [NO_x_^−^]: 4000 µM	*Escherichia coli* K-12 (KCTC 1116), *Pseudomonas aeruginosa* (KCTC 1636), *Staphylococcus aureus* (ATCC 12600)	Reduced viable cells, shifted gene expression (soxRS up, oxyR down), leads to nitric oxide accumulation, and alters bacterial cell morphology.	[130]
APPJ	Commercially purified water of pharmaceutical degree (Pharmacopoeia Europea, Ph. Eur. 9)	pH: 6.1, 6.1, 5.8 Ec: ~27 µS/cm [H_2_O_2_]: 4.1, 3.1, 0 mg/L [NO_2_^−^]: 3.3, 2.8, 1.4 mg/L [NO_3_^−^]: 11.2, 5.5, 5.4 mg/L For 10, 20, 30 min after plasma treatment, respectively	Lettuce (*Lactuca sativa* L.)	Better growth on day 7, but advantages faded by days 14, 21, and 28. No significant root system impact. Increased dry matter content.	[131]
APPJ	Commercially purified water of pharmaceutical degree (Pharmacopoeia Europea, Ph. Eur. 9)	pH: 6.1, 6.1, 5.8 [H_2_O_2_]: 4.1, 3.1, 0 mg/L [NO_2_^−^]: 3.3, 2.8, 1.4 mg/L [NO_3_^−^]: 11.2, 5.5, 5.4 mg/L For 10, 20, and 30 min after plasma treatment, respectively	Sweet pepper seeds (*Capsicum annuum*): ‘Bibic’ and ‘Bernita’	Height, weight, leaf count, interleaf nodes, and buds increased. Dry matter content rose. Yield increased.	[132]
Gliding arc plasma jet	Tap water, deionized water, distilled water, filtered water, and 0.9% saline	Plasma-treated tap water: pH: 5.56, 3.55, 3.06, 2.57 ± 0.09 ORP: 83, 193, 221, 250 ± 5.0 mV TDS: 50, 160, 330, 720 ± 5.0 ppm Ec: 70, 220, 470, 720 ± 5 μS/cm [H_2_O_2_]: 15.0, 80.3, 127.3, 111.7 mg/L [NO_2_^−^]: 25.0, 47.6, 48.6, 55.6 mg/L [NO_3_^−^]: 24.0, 53.3, 120.4, 69.3 mg/L Plasma-treated deionized water: pH: 4.00, 3.30, 2.95, 2.47 ± 0.09 ORP: 169, 208, 228, 239 ± 5.0 mV TDS: 40, 170, 450, 560 ± 5.0 ppm Ec: 50, 240, 310, 800 ± 5 μS/cm [H_2_O_2_]: 9.3, 76.5, 82.0, 160.3 mg/L [NO_2_^−^]: 9.2, 46.2, 45.8, 62.0 mg/L [NO_3_^−^]: 7.3, 47.4, 106.3, 69.3 mg/L After 1.0, 5.0, 30.0, and 60.0 min., respectively	*Escherichia coli* Oral keratinocyte cell cultures	Strong antimicrobial efficacy. Low cytotoxicity on oral keratinocytes.	[133]
APPJ	Acidic electrolyzed water	pH: ~2.3, 2.5, 1.95, 1.91 ORP: ~605, 608, 610, 855 mV Ec: ~1600, 2470, 2750, 3200 μS/cm [H_2_O_2_]: 75, 82, 75, 72 mg/L [NO_2_^−^]: ~85, 90, 90, 90 mg/L [NO_3_^−^]: 2.02, 75.28, 138.7 and 219.6 mg/L With plasma activation times of 3, 6, 10, and 15 min, respectively	*B. subtilis* (ATCC6633) and *E. coli* (ATCC8739)	Plasma-activated acidic electrolyzed water (PA-AEW) effectively kills *B. subtilis*, surpassing PAW and AEW.	[134]
Pinhole plasma jet	Deionized water	[H_2_O_2_]: 369.12 mg/L	*Escherichia coli* and *Colletotrichum gloeosporioides* in chili (*Capsicum annuum* L.)	Carbendazim and chlorpyrifos reduction. 100% fungal spore germination inhibition.	[135]
APPJ	Ultrapure milli-Q	pH: 6.5, 6.0 Ec: 99.0 ± 13.5, 177.3 ± 11.0 µS/cm ORP: 423.3 ± 11.7, 483.3 ± 12.6 mV TDS: 36.3 ± 2.1, 68.3 ± 5.5 ppm [O_3_]: 2.8 ± 0.1, 2.1 ± 0.3 mg/L [H_2_O_2_]: 0.5 ± 0.1, 1.5 ± 0.2 mg/L [NO_2_^−^]: 5.1 ± 0.4, 10.5 ± 1.1 mg/L [NO_3_^−^]: 34 ± 2.6, 53.9 ± 3.6 mg/L For 5- and 10-min plasma treatment, respectively	Pea (*Pisum sativum* L.) seeds	PAW-treated pea seeds had improved germination, growth, and biochemical traits. Wax removal, increased hydrophilicity, and enhanced antioxidant enzyme activity.	[136]
APPJ	Deionized water and deionized water +Mg^2+^	For PAW-Mg^2+^ pH: ~4.8, 5.2, 5.8, 6.0, 6.6 [NO_3_^−^]: ~25, 40, 40, 50, 60 mM For PAW pH: ~3.6, 3.2, 2.6, 2.2, 2.3 [NO_3_^−^]: ~20, 40, 80, 120, 120 mM For 3, 6, 10, 15, and 20 min plasma treatment, respectively	*Cryptococcus pseudolongus*	Magnesium-enriched PAW controls nitrate and exhibit antifungal activity against *C. pseudolongus*, with nitrate capture influencing its effectiveness.	[137]
APPJ	Ultrapure milli-Q	pH: ~2.5, 2.2, 2.0, 1.5, 1.5, 1.0 Ec: ~1, 3, 9, 22, 32, 39 mS/cm [NO_3_^−^]: ~200, 300, 500, 800, 900, 1200 mg/L [NO_2_^−^]: ~1500, 3000, 10,000, 14,000, 16,000, 16,000 µg/L [O_3_]: ~26, 24, 24, 17, 15, 14 mg/L [H_2_O_2_]: ~6, 4.5, 4.5, 3, 2.5, 2 mg/L For ORP 590, 630, 640, 700, 760, and 795 mV, respectively	*C. albicans*, *Citrus limon*	Reduced *C. albicans* by damaging cell membranes. Long-term fungicidal effects on *C. albicans* and *Citrus limon* spoilage fungi.	[138]
APPJ	Distilled water	pH: 6.50 ± 0.07 [O_3_]: 0.45 ± 0.01 mg/L [H_2_O_2_]: 8.75 ± 0.09 mg/L [NO_2_^−^]: 6.00 ± 0.06 mg/L [NO_3_^−^]: 46.00 ± 0.47 mg/L	Paddy seeds (*Oryza sativa* L.‘BRRIdhan 28’)	Enhanced rice seed germination, improved plant growth, defense mechanisms, enzymatic activity, protein, sugar content, and yielding.	[139]
APPJ	Reverse osmotic water	pH: 6.79 ± 0.18, 4.22 ± 0.07, 3.37 ± 0.06, 3.17 ± 0.06 Ec: 104.9 ± 22.55, 117.9 ± 2.80, 208.3 ± 6.51, 311.7 ± 12.01 µS/cm ORP: 370 ± 7.51, 465 ± 9.29, 534 ± 13.0, 554 ± 2.65 mV [H_2_O_2_]: ~ 78, 142, 115, 95 mg/L [NO_2_^−^]: 10.3 ± 0.58, 17.3 ± 1.15, 13.0 ± 1.00, 14.7 ± 0.58 mg/L [NO_3_^−^]: 13.8 ± 0.15, 29.8 ± 1.67, 33.3 ± 0.87, 42.7 ± 0.70 mg/L For 5, 10, 15, and 20 min plasma treatment, respectively	Water spinach (*Ipomoea aquatica*) seeds	Reduced Cd uptake in spinach but not Pb.	[140]
APPJ	Sterile deionized water	pH: 3.53, 3.24, 3.10, 3.01 [H_2_O_2_]: ~25, 50, 60, 55 µM [NO_x_]: ~0.9, 1.7, 2.5, 2.7 mM For 1, 3, 6, and 10 min by the soft plasma jet treatment, respectively.	*Aspergillus brasiliensis*	Damaged cell walls and reduced spore viability	[141]
APPJ	Deionized water	pH: 4.3 ± 0.3, 4.7 ± 0.2, 5.1 ± 0.3, 6.2 ± 0.4 [NO_x_]: 490.0 ± 53.7, 520.7 ± 71.6, 450.1 ± 69.5, 597.5 ± 53.4 M [NH_3_]: 2.1 ± 0.1, 2.6 ± 0.3, 2.9 ± 0.2, 4.9 ± 0.3 mg/L [H_2_O_2_]: 38.2 ± 5.0, 35.3 ± 3.1, 32.6 ± 6.5, 28.7 ± 4.2 M For PAW, Zn- PAW, Al- PAW, and Mg- PAW, respectively.	Maize plants (*Zea mays* L.)	PAW with added Mg, Al, or Zn ions accelerated germination and stem growth. Metal ions enhanced nitrogen reduction, leading to increased chlorophyll and protein content in plants.	[142]
Plasma torch	Deionized water	pH: ~3 [H_2_O_2_]: ~0.5 mg/L [NO_2_^−^]: ~47 mg/L [NO_3_^−^]: ~33 mg/L	*Arabidopsis thaliana*	PAW exposure rapidly increased Ca^2+^ in cells. Nitrates, nitrites, and hydrogen peroxide at PAW concentrations did not affect Ca^2+^ dynamics.	[143]
Plasma torch	Distilled water	pH: ~3 [H_2_O_2_]: 70 µM [NO_x_^−^]: 15 mM	Strawberry seeds	Improved germination	[144]
Plasma torch	Distilled water	pH: ~3 [H_2_O_2_]: 22.8 µM [NO_x_^−^]: 5.7 mM	Cotton, wheat, and strawberry seeds	Improved germination. Protection against fusarium and hyperthermia. PAW is better than commercial seed germination stimulators.	[145]
Corona discharge	Distilled water	pH: 3.3. [H_2_O_2_]: 4.5 ± 0.1 [NO_2_^−^]: 30.4 ± 0.9 [O_3_]: 0.3 ± 0.1 mg/L	Fresh rocket (*Eruca sativa*) leaves	Preserved nutrients, increased some vitamins and polyphenols. Decreased antioxidant and catalase activity, aiding storage quality.	[146]
Corona discharge	Distilled water	[H_2_O_2_]: 97.4 ± 3.2 mg/L [NO_3_^−^]: 391.1 ± 9.3 mg/L [O_3_]: <3.8 mg/L, [NO_2_^−^]: <10^−3^ mg/L of NO_2_^−^ mg/L	Wheat grains (*Triticum aestivum* L.), *Escherichia coli*, *Saccharomyces cerevisiae*	Improved germination, shoot growth, and decontamination of *E. coli* and *S. cerevisiae*.	[147]
Corona discharge	Distilled water	pH: 3.3 [H_2_O_2_]: 4.5 ± 0.1 mg/L [NO_2_^−^]: 30.4 ± 0.9 mg/L [O_3_]: 0.3 ± 0.1 mg/L	Fresh-cut *Eruca sativa*	Reduced radical scavenging activity over time but increased total phenols and glucosinolate percentage. PAW polyphenol extract showed no cytotoxicity and improved redox balance.	[148]
Gliding arc plasma	Tap water	pH: 3.4, 3.3 TDS: 90, 110 mg/L Ec: 190, 230 μS/cm [H_2_O_2_]: 0.01, 0.028 mM/L [NO_3_^−^]: 0.156, 0.176 mM/L For 5 and 15 min after plasma treatment, respectively	Maize (*Zea mays* L.) seeds	15 min treatment led to 100% seed germination and improved stem length, leaf width, collar diameter, chlorophyll content, and water uptake.	[149]
Gliding arc plasma	Distilled water	pH: ~4.2 ± 0.2, 3.7 ± 0.1, 3.3 ± 0.3 [H_2_O_2_]: ~6 ± 1, 7 ± 3, 12 ± 5 µM [NO_2_^−^]: ~1.9 ± 0.4, 2.4 ± 0.3, 2.9 ± 0.6 mM For 5, 10, and 20 min plasma treatment, respectively	Beetroot (*Beta vulgaris* ‘CYLINDRA’ ), carrot (*Daucus carota ‘*AFALON F1’) seeds	Improved germination in *Beta vulgaris* and had mixed effects on Daucus carota seeds compared to NaOCl, also altering fungal species composition.	[150]
Gliding arc plasma	Nutrient solution	[NO_3_^−^]: 191.9 ± 3.1, 189.8 ± 1.7, 191.6 ± 1.6 [NO_2_^−^]: 18.8 ± 1.9, 18.6 ± 0.9, 18.7 ± 1.1, For one-time DBD processing and multiple processing, respectively	*Ocimum basilicum* L.	Boosted basil growth, enriched aroma compounds, and reduced algae.	[151]
Gliding arc plasma	Deionized water	pH: 4.69 ± 0.08, 4.48 ± 0.05, 4.37 ± 0.07, 4.29 ± 0.06 T: 16.5 ± 0.04, 16.59 ± 0.06, 16.62 ± 0.03, 16.64 ± 0.04 °C Ec: 50.2 ± 1.60; 60.2 ± 1.20, 70.6 ± 1.80, 76.8 ± 1.60 µS/cm [NO_2_^−^]: 0.49 ± 0.04, 0.60 ± 0.03, 0.80 ± 0.06, 0.86 ± 0.04 mg/L [NO_3_^−^]: 0.49 ± 0.14, 4.84 ± 0.12, 6.69 ± 0.16, 7.20 ± 0.12 mg/L [NH_3_]: 1.11 ± 0.07, 2.68 ± 0.09, 3.01 ± 0.07, 3.40 ± 0.08 mg/L for 5, 10, 15, 20. GAD treatment, respectively	Phapar (*Fagopyrum esculentum*), barley (*Hordeum vulgare*), mustard (*Brassica nigra*), and rayo (*Brassica juncea*)	Improved seed germination, uniformity, daily germination, increased water uptake, root/shoot length, and seed vigor.	[152]
RF discharge	Distilled water	pH: 3.0 [H_2_O_2_]: 100 ppm	Flour from Noui Khuea brown rice (*Oryza sativa* L.) conjugated with three different phenolic compounds—gallic acid, sinapic acid, and crude Mon-pu (*Glochidion wallichianum* Muell Arg) extract	Gallic acid boosts 1,1-diphenyl-2-picrylhydrazyl removal in starch, especially when PAW-synthesized, regardless of ultrasound. Complexation index, resistant starch lower than crude Mon-pu extract complex.	[153]
Direct discharge	Distilled water	pH: 3.3 [O_3_]: 0.3 ± 0.1 mg/L [H_2_O_2_]: 4.5 ± 0.1 mg/L [NO_2_^−^]: 30.4 ± 0.9 mg/L	*Eruca sativa* Mill	β-sitosterol and campesterol decreased, while β-carotene, luteolin, and chlorophyll b increased after 2-min PAW exposure. Chlorophyll content reduced at 20 min.	[154]
Direct discharge	Distilled water	pH: 3.3 [O_3_]: 0.3 ± 0.1 mg/L [H_2_O_2_]: 4.5 ± 0.1 mg/L [NO_2_^−^]: 30.4 ± 0.9 mg/L	*Eruca sativa*	Reduced bacteria by 1.7–3 Log CFU/g with minor quality changes, surpassing the antibacterial effect of hypochlorite.	[155]
Transient spark discharge	Tap water	pH: 7.5 [H_2_O_2_]: 0.5 ± 0.1 mM [NO_2_^−^]: 0.6 ± 0.1 mM [NO_3_^−^]: 1.7 ± 0.3 mM	Corns of maize (*Zea mays* L.) (hybrid Bielik)	Enhanced seedling growth. It affected chlorophyll, carotenoid levels, and leaf arsenic accumulation, not root accumulation.	[156]
Glow discharge	Tap water	[H_2_O_2_]: ~0.33, 0.64, 1.00 mM [NO_2_^−^]: ~0.93, 0.59, 0.95 mM [NO_3_^−^]: ~2.46, 1.40, 2.34 mM For transient spark, PAW of glow discharge at activation time 1 min and 2 min, respectively.	Dried barley (*Hordeum vulgare* L. ‘Kangoo’) grains and pea (*Pisum sativum* L. ‘Eso’) seeds	Enhanced pea growth and amylase activity without harm, while barley suffered DNA damage, growth reduction, and oxidative stress.	[157]
Glow discharge.	Re-distilled water and the inorganic salt concentration (ammonium nitrate, NH_4_NO_3_) in the solution turned out to be 0.50% (m/w)	−	*Dickeya solani* IFB0099, *Pectobacterium atrosepticum* IFB5103 strains	Bactericidal effects on *Dickeya solani* and *Pectobacterium atrosepticum* within 24 h.	[158]
Transient spark	Tap water	pH: ~ 7.5 [H_2_O_2_]: ~ 0.42 mM [NO_3_^−^]: ~ 0.85 mM	Lettuce (*Lactuca sativa* L. var. *capitata* ‘Král máje I’)	PAW-irrigated lettuce had similar dry weight but higher pigment content, photosynthetic rate, and lower antioxidant enzyme activity compared to H_2_O_2_ + NO_3_^−^ irrigation.	[159]
Glow discharge	Tap water	−	*Poa pratensis, Lolium perenne*	Decreased fungal diseases, improved turf density and overwintering. Synergistic effect with bio-stimulant.	[160]
Spark discharge	Deionized water	pH: 6.7, 6.4, 6.0, 5.3, 3.7 [H_2_O_2_]: ~60, 75, 100, 150, 240 µM [NO_2_^−^]: 25, 30, 50, 75, 230 µM For 3, 6, 9,12, and 15 min plasma treatment, respectively.	Black gram (*Vigna mungo* L.)	Increased H_2_O_2_ and ROS levels in seeds, leaves, and roots. Elevated catalase levels linked to VmCAT gene activation.	[161]
Glow discharge	80 mM L-phenylalanine	[H_2_O_2_]: 282 µM [NO_2_^−^]: 4.4 µM [NO_3_^−^]: 520 µM after 4 min plasma treatment	*Raphanus* sp., *E. coli* O1:K1:H7	Seedlings stimulation and antibacterial effect.	[162]
T-shaped reactor	Distilled water	pH: 4.6; 4.4; 5.1; 4.1; 5.1; 3.9; 5.5; 3.8 [NO_3_^−^]: 30, 40, 14, 49, 15, 55, 8, 68 ± 10% mg/L [H_2_O_2_]: 7, 8, 3, 13, 5, 14, 1, 22 ± 5% mg/L During treatment in the modes: Frequency: 150, 250, 60, 250, 60, 250, 250, 250 Hz; Duration: 2, 2, 2, 2, 1.5, 1.5, 2, 2 ms; Airflow: 1, 1, 1, 1, 1, 1, 2.5, 2.5 L/min; Liquid flow: 10, 10, 20, 3.33, 25, 3.33; 85, 4 L/min, respectively	Wheat (*Triticum aestivum*) grains	Reduced mitotic activity in wheat sprouts’ cells. Genotoxicity tied to H_2_O_2_, NO_3_^−^ levels. Positive effect on wheat germination	[163]
Underwater discharge	Distilled water	pH: ~3.91 ± 0.03, [O_3_]: ~0.25 ± 0.01 mg/L [H_2_O_2_]: ~7.50 ± 0.05 mg/L [NO_2_^−^]: ~4.59 ± 0.04 mg/L [NO_3_^−^]: ~26.58 ± 0.25 mg/L	Potato (*Solanum tuberosum* L.)	PAW spraying on potato foliage enhanced growth, enzyme activity, protein and sugar content, and tuber yield.	[164]
Underwater discharge	Distilled water with KH_2_PO_4_	[H_2_O_2_]: 100 μM	Spring spelled seeds, spring rye seeds	Improved germination, improved root development.	[165]
(1) Underwater discharge. (2) Plasma torch	(1) Distilled water + KCl (2) Distilled water	(1) pH: 8.3 [H_2_O_2_]: 7.12 mM [NO_x_^−^]: 22.05 mM (2) pH: 4.5 [H_2_O_2_]: 0.11 mM [NO_x_^−^]: 87 mM	Apple tree	Increased primary nutrient content in fruits and leaves. Ca in the fruits increased significantly.	[166]
Underwater discharge	Distilled water + KCl	pH: 8.3 [H_2_O_2_]: 7.12 mM [NO_x_^−^]: 22.05 mM	Sorghum and barley seeds, strawberry crops	Improved germination and crop development. Increased drought resistance.	[167]

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
