# Peer review of "Advancements in Plasma Agriculture: A Review of Recent Studies"

_ijms, 2023, doi:10.3390/ijms242015093_

Round 1
Reviewer 1 Report
Review of an article entitled Advancements in Plasma Agriculture: A Review of Recent Studies. An interesting article based on a wide range of recent literature. The layout of the manuscript is correct and clear.
The authors have not shied away from some errors in the text:
Line 73 - shouldn't there be a marker next to the formulae given in brackets that it is a radical?
Figure 3 - I would change where a), b), ... are marked; perhaps in the top left-hand corner to make it more visible?; it seems that the current notation may be a little confusing
Table 1 - in the second column, please standardise the upper/lower case notation; in addition, such a large table should be transferred between pages with the header and signature
Table 2 - check notations, e.g. PH instead of pH in the first row in the third column; there are no spaces between the values and the +/- sign in the first row and there are already spaces in the second row; standardise the upper/lower case notation in column 2
Please check the superscript and subscript notation in the text (e.g. line 435 and many others)
line 487-488 - was variation in oxidation levels of elements taken into account? did this affect the process?
lines 593-594 - Sodium Hypochlorite please write as NaClO
General comment - I suggest that where the sentence subject is the citation number it would be better to cite the research group leader
Author Response
We thank the reviewer for his kind review. The following changes have been made to the manuscript.
Comment 1.1
Line 73 - shouldn't there be a marker next to the formulae given in brackets that it is a radical?
Reply 1.1
Corrected
Comment 1.2
Figure 3 - I would change where a), b), ... are marked; perhaps in the top left-hand corner to make it more visible?; it seems that the current notation may be a little confusing
Reply 1.2
Changes applied
Comment 1.3
Table 1 - in the second column, please standardise the upper/lower case notation; in addition, such a large table should be transferred between pages with the header and signature
Reply 1.3
Corrected
Comment 1.4
Table 2 - check notations, e.g. PH instead of pH in the first row in the third column; there are no spaces between the values and the +/- sign in the first row and there are already spaces in the second row; standardise the upper/lower case notation in column 2
Reply 1.4
Corrected
Comment 1.5
Please check the superscript and subscript notation in the text (e.g. line 435 and many others)
Reply 1.5
Corrected
Comment 1.6
line 487-488 - was variation in oxidation levels of elements taken into account? did this affect the process?
Reply 1.6
The paper [Lamichhane, P. et al. Low-Temperature Plasma-Assisted Nitrogen Fixation for Corn Plant Growth and Development. IJMS 2021, 22, 5360, doi:10.3390/ijms22105360] comprehensively discussed the chemical processes in water and the role of the metals oxidation levels. Metals immersed in water undergo oxidation, resulting in the generation of metal ions. These metal ions subsequently convert nitrite (NO2−) and nitrate (NO3−) species into their corresponding metal nitrates and nitrites, respectively. Concurrently, the electrons from the metal species leads to the reduction of hydrogen ions (H+) to hydrogen (H), thereby inducing a substantial increase in the solution's pH level. Hydrogen facilitates the reduction of nitrogen to yield ammonia (NH3). As a consequence, the rates of ammonia synthesis and the degree of pH elevation within various activated waters follow the sequence: Zn-PAW < Al-PAW < Mg-PAW. At restricted concentrations, metal ions such as magnesium ions (Mg2+), aluminum ions (Al3+), and zinc ions (Zn2+) serve advantageous functions in plant physiology.
This explanation has been included in the manuscript.
Comment 1.7
lines 593-594 - Sodium Hypochlorite please write as NaClO
Reply 1.7
Corrected
Comment 1.8
General comment - I suggest that where the sentence subject is the citation number it would be better to cite the research group leader
Reply 1.8
For references, we used the template provided by the publisher for the reference formatting software. We are concerned that deviations from the template may lead to a requirement from the editors to correct this at the final stage of editing. If this comment is critical to the approval of the article, we will make the change in the next round of revision.
Reviewer 2 Report
Reviewer comment: Thanks for inviting me to review this review article, " Advancements in Plasma Agriculture: A Review of Recent Studies". I found that the presentation of this work is good, which would be helpful for the possible reader of the "International Journal of Molecular Sciences". Therefore, I recommend publishing this review article in the "International Journal of Molecular Sciences" but only after a manuscript revision. Please find my comments below.
1. The introduction of this review is poor. Therefore, I suggest showing a wide application of plasma and discussing it a little bit in the introduction.
2. Please add some OES data and discuss the plasma species generated by both gas-phase plasma and liquid-phase plasma. For your information, see the references: Arabian Journal of Chemistry: Volume 16, Issue 10, October 2023, 105174 (https://doi.org/10.1016/j.arabjc.2023.105174); Surfaces and Interfaces: Volume 35, December 2022, 102462 (https://doi.org/10.1016/j.surfin.2022.102462); Journal of Environmental Chemical Engineering: Volume 9, Issue 4, August 2021, 105780 (https://doi.org/10.1016/j.jece.2021.105780); and Water Research: Volume 224, 1 October 2022, 119107 (https://doi.org/10.1016/j.watres.2022.119107).
Author Response
We thank the reviewer for his kind review. The following changes have been made to the manuscript.
Comment 2.1
- The introduction of this review is poor. Therefore, I suggest showing a wide application of plasma and discussing it a little bit in the introduction.
Reply 2.1
Into the Introduction section we include brief insight to the wide plasma applications to not conflict with the same information from section 2.
“The unique properties of physical plasma, including its ability to operate at atmospheric pressure, make it an attractive technology for numerous scientific and industrial applications, ranging from medicine and agriculture to electronics and materials science. For example, this technology has proven itself to be a simple and low-cost approach for nanoparticles synthesis [10.1016/j.arabjc.2023.105174], or to be an effective surface modification agent to produce superhydrophobic and superoleophilic films for oil-water separation, self-cleaning [10.1016/j.surfin.2022.102462]. In medicine, plasma is widely used to solve problems of creating biocompatible materials, orthopedics [10.1530/EOR-22-0106], oncology, dentistry [10.1007/s11090-023-10380-5], dermatology [10.1615/PlasmaMed.2023049359], etc.”
Comment 2.2
- Please add some OES data and discuss the plasma species generated by both gas-phase plasma and liquid-phase plasma. For your information, see the references: Arabian Journal of Chemistry: Volume 16, Issue 10, October 2023, 105174 (https://doi.org/10.1016/j.arabjc.2023.105174); Surfaces and Interfaces: Volume 35, December 2022, 102462 (https://doi.org/10.1016/j.surfin.2022.102462); Journal of Environmental Chemical Engineering: Volume 9, Issue 4, August 2021, 105780 (https://doi.org/10.1016/j.jece.2021.105780); and Water Research: Volume 224, 1 October 2022, 119107 (https://doi.org/10.1016/j.watres.2022.119107).
Reply 2.2
Into the section 2 we included the discussion of OES analysis in plasma-gas-liquid interaction.
“In practice, determining the optimal parameters for plasma-liquid interaction can be quite challenging. These parameters include the applied voltage, discharge gas flow rate, treatment duration, distance of the electrode from the solution surface, and the potential concentration of precursors or reducing agents. In the gas and liquid phases, free electrons receive energy from the electric field created between electrodes. This energy leads to the generation of relatively high-energy electrons. These high-energy electrons can then interact with various components, such as oxygen, water molecules, and other ionic compounds (1-3), triggering a range of physical and chemical reactions. These processes have the potential to produce reactive chemical species like OH•, O•, H•, ONOO−, NO•, H2O2, and ultraviolet (UV) radiation. These species can be detected through the emission of photons, which can be analyzed using optical emission spectroscopy (OES).
N2 + e– → •N + •N + e– (1)
O2 + e– → •O + •O + e– (2)
H2O + e– → •H + •OH + e– (3)
Nitrogen in emission spectrum can be relatively clearly determined from the N2 second positive system (C3Πu−B3Πg) at wavelength from 315 to 433 nm. [10.1016/j.surfin.2022.102462, 10.1016/j.jece.2021.105780]. The presence of O2 in the discharge zone can be indicated by the NO group peaks from 214 to 270 nm. The •HO radicals are confirmed by the peak at about 296 nm or 310 nm, the oxygen radical at 284 nm [10.1016/j.watres.2022.119107]. The emission peaks corresponding to •OH and •O may be attributed to the fragmentation of H2O molecules. These H2O molecules can either diffuse in from the surrounding ambient air or evaporate from the liquid's surface.
To achieve the required biological effect, it is also important to analyze nanoparticles (NP) that can form in plasma-activated water as a result of erosion of electrodes or other metal objects immersed in the liquid. Transmission electron microscopy can be used to obtain surface morphology and particle size distribution, whereas elemental characterization of NP is often obtained with energy dispersive spectroscopy. The stability test of NP and elemental characterization of PAW are analyzed, for example, using ultraviolet–visible spectroscopy (UV–Vis). Thus, the UV–Vis absorption peaks at around 410 nm corresponds to the silver NP formation [10.1016/j.arabjc.2023.105174].”
Round 2
Reviewer 2 Report
The authors made significant improvements to the article in response to the reviewers' criticisms. In light of this, I suggest accepting.